# On Calibrating Diffusion Probabilistic Models

**Tianyu Pang**[†1]**, Cheng Lu**[2]**, Chao Du**[1]**, Min Lin**[1]**, Shuicheng Yan**[1]**, Zhijie Deng**[†3]
[1]Sea AI Lab, Singapore
[2]Department of Computer Science, Tsinghua University
[3]Qing Yuan Research Institute, Shanghai Jiao Tong University
{tianyupang, duchao, linmin, yansc}@sea.com;
lucheng.lc15@gmail.com; zhijied@sjtu.edu.cn

## Abstract

Recently, diffusion probabilistic models (DPMs) have achieved promising results in diverse generative tasks. A typical DPM framework includes a forward process that gradually diffuses the data distribution and a reverse process that recovers the data distribution from time-dependent data scores. In this work, we observe that the stochastic reverse process of data scores is a martingale, from which concentration bounds and the optional stopping theorem for data scores can be derived. Then, we discover a simple way for *calibrating* an arbitrary pretrained DPM, with which the score matching loss can be reduced and the lower bounds of model likelihood can consequently be increased. We provide general calibration guidelines under various model parametrizations. Our calibration method is performed only once and the resulting models can be used repeatedly for sampling. We conduct experiments on multiple datasets to empirically validate our proposal. Our code is available at https://github.com/thudzj/Calibrated-DPMs.

## 1 Introduction

In the past few years, denoising diffusion probabilistic modeling [17, 40] and score-based Langevin dynamics [42, 43] have demonstrated appealing results on generating images. Later, Song et al. [46] unify these two generative learning mechanisms through stochastic/ordinary differential equations (SDEs/ODEs). In the following we refer to this unified model family as diffusion probabilistic models (DPMs). The emerging success of DPMs has attracted broad interest in downstream applications, including image generation [10, 22, 48], shape generation [4], video generation [18, 19], super-resolution [35], speech synthesis [5], graph generation [51], textual inversion [13, 34], improving adversarial robustness [50], and text-to-image large models [32, 33], just to name a few.

A typical framework of DPMs involves a *forward* process gradually diffusing the data distribution $q_0(x_0)$ towards a noise distribution $q_T(x_T)$. The transition probability for $t \in [0, T]$ is a conditional Gaussian distribution $q_{0t}(x_t|x_0) = \mathcal{N}(x_t|\alpha_t x_0, \sigma_t^2 \mathbf{I})$, where $\alpha_t, \sigma_t \in \mathbb{R}^+$. Song et al. [46] show that there exist *reverse* SDE/ODE processes starting from $q_T(x_T)$ and sharing the same marginal distributions $q_t(x_t)$ as the forward process. The only unknown term in the reverse processes is the data score $\nabla_{x_t} \log q_t(x_t)$, which can be approximated by a time-dependent score model $s_\theta^t(x_t)$ (or with other model parametrizations). $s_\theta^t(x_t)$ is typically learned via score matching (SM) [20].

In this work, we observe that the stochastic process of the scaled data score $\alpha_t \nabla_{x_t} \log q_t(x_t)$ is a *martingale* w.r.t. the reverse-time process of $x_t$ from $T$ to $0$, where the timestep $t$ can be either continuous or discrete. Along the reverse-time sampling path, this martingale property leads to concentration bounds for scaled data scores. Moreover, a martingale satisfies the optional stopping theorem that the expected value at a stopping time is equal to its initial expected value.

---

[†]Corresponding authors.

Based on the martingale property of data scores, for any $t \in [0, T]$ and any pretrained score model $s_\theta^t(x_t)$ (or with other model parametrizations), we can *calibrate* the model by subtracting its expectation over $q_t(x_t)$, i.e., $\mathbb{E}_{q_t(x_t)}[s_\theta^t(x_t)]$. We formally demonstrate that the calibrated score model $s_\theta^t(x_t) - \mathbb{E}_{q_t(x_t)}[s_\theta^t(x_t)]$ achieves lower values of SM objectives. By the connections between SM objectives and model likelihood of the SDE process [23, 45] or the ODE process [28], the calibrated score model has higher evidence lower bounds. Similar conclusions also hold for the conditional case, in which we calibrate a conditional score model $s_\theta^t(x_t, y)$ by subtracting its conditional expectation $\mathbb{E}_{q_t(x_t|y)}[s_\theta^t(x_t, y)]$.

In practice, $\mathbb{E}_{q_t(x_t)}[s_\theta^t(x_t)]$ or $\mathbb{E}_{q_t(x_t|y)}[s_\theta^t(x_t, y)]$ can be approximated using noisy training data when the score model has been pretrained. We can also utilize an auxiliary shallow model to estimate these expectations dynamically during pretraining. When we do not have access to training data, we could calculate the expectations using data generated from $s_\theta^t(x_t)$ or $s_\theta^t(x_t, y)$. In experiments, we evaluate our calibration tricks on the CIFAR-10 [25] and CelebA $64 \times 64$ [27] datasets, reporting the FID scores [16]. We also provide insightful visualization results on the AFHQv2 [7], FFHQ [21] and ImageNet [9] at $64 \times 64$ resolution.

## 2 Diffusion probabilistic models

In this section, we briefly review the notations and training paradigms used in diffusion probabilistic models (DPMs). While recent works develop DPMs based on general corruptions [2, 8], we mainly focus on conventional Gaussian-based DPMs.

### 2.1 Forward and reverse processes

We consider a $k$-dimensional random variable $x \in \mathbb{R}^k$ and define a *forward* diffusion process on $x$ as $\{x_t\}_{t \in [0,T]}$ with $T > 0$, which satisfies $\forall t \in [0, T]$,

$$x_0 \sim q_0(x_0), \qquad q_{0t}(x_t|x_0) = \mathcal{N}(x_t|\alpha_t x_0, \sigma_t^2 \mathbf{I}). \tag{1}$$

Here $q_0(x_0)$ is the data distribution; $\alpha_t$ and $\sigma_t$ are two positive real-valued functions that are differentiable w.r.t. $t$ with bounded derivatives. Let $q_t(x_t) = \int q_{0t}(x_t|x_0)q_0(x_0)dx_0$ be the marginal distribution of $x_t$. The schedules of $\alpha_t, \sigma_t^2$ need to ensure that $q_T(x_T) \approx \mathcal{N}(x_T|0, \widetilde{\sigma}^2 \mathbf{I})$ for some $\widetilde{\sigma}$. Kingma et al. [23] prove that there exists a stochastic differential equation (SDE) satisfying the forward transition distribution in Eq. (1), and this SDE can be written as

$$dx_t = f(t)x_t dt + g(t)d\omega_t, \tag{2}$$

where $\omega_t \in \mathbb{R}^k$ is the standard Wiener process, $f(t) = \frac{d \log \alpha_t}{dt}$, and $g(t)^2 = \frac{d\sigma_t^2}{dt} - 2\frac{d \log \alpha_t}{dt}\sigma_t^2$. Song et al. [46] demonstrate that the forward SDE in Eq. (2) corresponds to a *reverse* SDE constructed as

$$dx_t = \left[f(t)x_t - g(t)^2 \nabla_{x_t} \log q_t(x_t)\right] dt + g(t)d\overline{\omega}_t, \tag{3}$$

where $\overline{\omega}_t \in \mathbb{R}^k$ is the standard Wiener process in reverse time. Starting from $q_T(x_T)$, the marginal distribution of the reverse SDE process is also $q_t(x_t)$ for $t \in [0, T]$. There also exists a deterministic process described by an ordinary differential equation (ODE) as

$$\frac{dx_t}{dt} = f(t)x_t - \frac{1}{2}g(t)^2 \nabla_{x_t} \log q_t(x_t), \tag{4}$$

which starts from $q_T(x_T)$ and shares the same marginal distribution $q_t(x_t)$ as the reverse SDE in Eq. (3). Moreover, let $q_{0t}(x_0|x_t) = \frac{q_{0t}(x_t|x_0)q_0(x_0)}{q_t(x_t)}$ and by Tweedie's formula [12], we know that $\alpha_t \mathbb{E}_{q_{0t}(x_0|x_t)}[x_0] = x_t + \sigma_t^2 \nabla_{x_t} \log q_t(x_t)$.

### 2.2 Training paradigm of DPMs

To estimate the data score $\nabla_{x_t} \log q_t(x_t)$ at timestep $t$, a score-based model $s_\theta^t(x_t)$ [46] with shared parameters $\theta$ is trained to minimize the score matching (SM) objective [20] as

$$\mathcal{J}_{\text{SM}}^t(\theta) \triangleq \frac{1}{2}\mathbb{E}_{q_t(x_t)}\left[\|s_\theta^t(x_t) - \nabla_{x_t} \log q_t(x_t)\|_2^2\right]. \tag{5}$$

To eliminate the intractable computation of $\nabla_{x_t} \log q_t(x_t)$, denoising score matching (DSM) [49] transforms $\mathcal{J}_{\text{SM}}^t(\theta)$ into $\mathcal{J}_{\text{DSM}}^t(\theta) \triangleq \frac{1}{2}\mathbb{E}_{q_0(x_0),q(\epsilon)}\left[\left\|s_\theta^t(x_t) + \frac{\epsilon}{\sigma_t}\right\|_2^2\right]$, where $x_t = \alpha_t x_0 + \sigma_t \epsilon$ and

$q(\epsilon) = \mathcal{N}(\epsilon|\mathbf{0}, \mathbf{I})$ is a standard Gaussian distribution. Under mild boundary conditions, we know $\mathcal{J}_{\text{SM}}^t(\theta)$ and $\mathcal{J}_{\text{DSM}}^t(\theta)$ is equivalent up to a constant, i.e., $\mathcal{J}_{\text{SM}}^t(\theta) = \mathcal{J}_{\text{DSM}}^t(\theta) + C^t$ and $C^t$ is a constant independent of the model parameters $\theta$. Other SM variants [31, 44] are also applicable here. The total SM objective for training is a weighted sum of $\mathcal{J}_{\text{SM}}^t(\theta)$ across $t \in [0, T]$, defined as $\mathcal{J}_{\text{SM}}(\theta; \lambda(t)) \triangleq \int_0^T \lambda(t) \mathcal{J}_{\text{SM}}^t(\theta) dt$, where $\lambda(t)$ is a positive weighting function. Similarly, the total DSM objective is $\mathcal{J}_{\text{DSM}}(\theta; \lambda(t)) \triangleq \int_0^T \lambda(t) \mathcal{J}_{\text{DSM}}^t(\theta) dt$. The training objectives under other model parametrizations such as noise prediction $\boldsymbol{\epsilon}_\theta^t(x_t)$ [17, 33], data prediction $\boldsymbol{x}_\theta^t(x_t)$ [23, 32], and velocity prediction $\boldsymbol{v}_\theta^t(x_t)$ [18, 38] are recapped in Appendix B.1.

### 2.3 Likelihood of DPMs

Suppose that the reverse processes start from a tractable prior $p_T(x_T) = \mathcal{N}(x_T|0, \widetilde{\sigma}^2 \mathbf{I})$. We can approximate the reverse-time SDE process by substituting $\nabla_{x_t} \log q_t(x_t)$ with $\boldsymbol{s}_\theta^t(x_t)$ in Eq. (3) as $dx_t = \left[ f(t)x_t - g(t)^2 \boldsymbol{s}_\theta^t(x_t) \right] dt + g(t) d\overline{\omega}_t$, which induces the marginal distribution $p_t^{\text{SDE}}(x_t; \theta)$ for $t \in [0, T]$. In particular, at $t = 0$, the KL divergence between $q_0(x_0)$ and $p_0^{\text{SDE}}(x_0; \theta)$ can be bounded by the total SM objective $\mathcal{J}_{\text{SM}}(\theta; g(t)^2)$ with the weighing function of $g(t)^2$, as stated below:

**Lemma 1.** *(Proof in Song et al. [45]) Let $q_t(x_t)$ be constructed from the forward process in Eq. (2). Then under regularity conditions, we have $\mathcal{D}_{KL}\left(q_0 \| p_0^{SDE}(\theta)\right) \leq \mathcal{J}_{SM}(\theta; g(t)^2) + \mathcal{D}_{KL}(q_T \| p_T)$.*

Here $\mathcal{D}_{\text{KL}}(q_T \| p_T)$ is the prior loss independent of $\theta$. Similarly, we approximate the reverse-time ODE process by substituting $\nabla_{x_t} \log q_t(x_t)$ with $\boldsymbol{s}_\theta^t(x_t)$ in Eq. (4) as $\frac{dx_t}{dt} = f(t)x_t - \frac{1}{2}g(t)^2 \boldsymbol{s}_\theta^t(x_t)$, which induces the marginal distribution $p_t^{\text{ODE}}(x_t; \theta)$ for $t \in [0, T]$. By the instantaneous change of variables formula [6], we have $\frac{\log p_t^{\text{ODE}}(x_t; \theta)}{dt} = -\mathbf{tr}\left(\nabla_{x_t}\left(f(t)x_t - \frac{1}{2}g(t)^2 \boldsymbol{s}_\theta^t(x_t)\right)\right)$, where $\mathbf{tr}(\cdot)$ denotes the trace of a matrix. Integrating change in $\log p_t^{\text{ODE}}(x_t; \theta)$ from $t = 0$ to $T$ can give the value of $\log p_T(x_T) - \log p_0^{\text{ODE}}(x_0; \theta)$, but requires tracking the path from $x_0$ to $x_T$. On the other hand, at $t = 0$, the KL divergence between $q_0(x_0)$ and $p_0^{\text{ODE}}(x_0; \theta)$ can be decomposed:

**Lemma 2.** *(Proof in Lu et al. [28]) Let $q_t(x_t)$ be constructed from the forward process in Eq. (2). Then under regularity conditions, we have $\mathcal{D}_{KL}\left(q_0 \| p_0^{ODE}(\theta)\right) = \mathcal{J}_{SM}(\theta; g(t)^2) + \mathcal{D}_{KL}(q_T \| p_T) + \mathcal{J}_{Diff}(\theta)$, where the term $\mathcal{J}_{Diff}(\theta)$ measures the difference between $\boldsymbol{s}_\theta^t(x_t)$ and $\nabla_{x_t} \log p_t^{ODE}(x_t; \theta)$.*

Directly computing $\mathcal{J}_{\text{Diff}}(\theta)$ is intractable due to the term $\nabla_{x_t} \log p_t^{\text{ODE}}(x_t; \theta)$, nevertheless, we could bound $\mathcal{J}_{\text{Diff}}(\theta)$ via bounding high-order SM objectives [28].

## 3 Calibrating pretrained DPMs

In this section we begin with deriving the relationship between data scores at different timesteps, which leads us to a straightforward method for calibrating any pretrained DPMs. We investigate further how the dataset bias of finite samples prevents empirical learning from achieving calibration.

### 3.1 The stochastic process of data score

According to Kingma et al. [23], the form of the forward process in Eq. (1) can be generalized to any two timesteps $0 \leq s < t \leq T$. Then, the transition probability from $x_s$ to $x_t$ is written as $q_{st}(x_t|x_s) = \mathcal{N}\left(x_t \middle| \alpha_{t|s}x_s, \sigma_{t|s}^2 \mathbf{I}\right)$, where $\alpha_{t|s} = \frac{\alpha_t}{\alpha_s}$ and $\sigma_{t|s}^2 = \sigma_t^2 - \alpha_{t|s}^2 \sigma_s^2$. Here the marginal distribution satisfies $q_t(x_t) = \int q_{st}(x_t|x_s) q_s(x_s) dx_s$. We can generally derive the connection between data scores $\nabla_{x_t} \log q_t(x_t)$ and $\nabla_{x_s} \log q_s(x_s)$ as stated below:

**Theorem 1.** *(Proof in Appendix A.1) Let $q_t(x_t)$ be constructed from the forward process in Eq. (2). Then under some regularity conditions, we have $\forall 0 \leq s < t \leq T$,*

$$\alpha_t \nabla_{x_t} \log q_t(x_t) = \mathbb{E}_{q_{st}(x_s|x_t)}\left[\alpha_s \nabla_{x_s} \log q_s(x_s)\right], \tag{6}$$

*where $q_{st}(x_s|x_t) = \frac{q_{st}(x_t|x_s) q_s(x_s)}{q_t(x_t)}$ is the transition probability from $x_t$ to $x_s$.*

Theorem 1 indicates that the stochastic process of $\alpha_t \nabla_{x_t} \log q_t(x_t)$ is a *martingale* w.r.t. the reverse-time process of $x_t$ from timestep $T$ to 0. From the optional stopping theorem [14], the expected

value of a martingale at a stopping time is equal to its initial expected value $\mathbb{E}_{q_0(x_0)}[\nabla_{x_0} \log q_0(x_0)]$. It is known that, under a mild boundary condition on $q_0(x_0)$, there is $\mathbb{E}_{q_0(x_0)}[\nabla_{x_0} \log q_0(x_0)] = 0$ (proof is recapped in Appendix A.2). Consequently, as to the stochastic process, the martingale property results in $\mathbb{E}_{q_t(x_t)}[\nabla_{x_t} \log q_t(x_t)] = 0$ for $\forall t \in [0, T]$. Moreover, the martingale property of the (scaled) data score $\alpha_t \nabla_{x_t} \log q_t(x_t)$ leads to concentration bounds using Azuma's inequality and Doob's martingale inequality as derived in Appendix A.3. Although we do not use these concentration bounds further in this paper, there are other concurrent works that use roughly similar concentration bounds in diffusion models, such as proving consistency [47] or justifying trajectory retrieval [52].

## 3.2 A simple calibration trick

Given a pretrained model $s_\theta^t(x_t)$ in practice, there is usually $\mathbb{E}_{q_t(x_t)}[s_\theta^t(x_t)] \neq 0$, despite the fact that the expect data score is zero as $\mathbb{E}_{q_t(x_t)}[\nabla_{x_t} \log q_t(x_t)] = 0$. This motivates us to calibrate $s_\theta^t(x_t)$ to $s_\theta^t(x_t) - \eta_t$, where $\eta_t$ is a time-dependent calibration term that is independent of any particular input $x_t$. The calibrated SM objective is written as follows:

$$
\begin{aligned}
\mathcal{J}_{\text{SM}}^t(\theta, \eta_t) &\triangleq \frac{1}{2}\mathbb{E}_{q_t(x_t)}\left[\|s_\theta^t(x_t) - \eta_t - \nabla_{x_t} \log q_t(x_t)\|_2^2\right] \\
&= \mathcal{J}_{\text{SM}}^t(\theta) - \mathbb{E}_{q_t(x_t)}\left[s_\theta^t(x_t)\right]^\top \eta_t + \frac{1}{2}\|\eta_t\|_2^2,
\end{aligned}
\tag{7}
$$

where the second equation holds after the results of $\mathbb{E}_{q_t(x_t)}[\nabla_{x_t} \log q_t(x_t)] = 0$, and there is $\mathcal{J}_{\text{SM}}^t(\theta, 0) = \mathcal{J}_{\text{SM}}^t(\theta)$ specifically when $\eta_t = 0$. Note that the orange part in Eq. (7) is a quadratic function w.r.t. $\eta_t$. We look for the optimal $\eta_t^* = \arg\min_{\eta_t} \mathcal{J}_{\text{SM}}^t(\theta, \eta_t)$ that minimizes the calibrated SM objective, from which we can derive

$$
\eta_t^* = \mathbb{E}_{q_t(x_t)}\left[s_\theta^t(x_t)\right].
\tag{8}
$$

After taking $\eta_t^*$ into $\mathcal{J}_{\text{SM}}^t(\theta, \eta_t)$, we have

$$
\mathcal{J}_{\text{SM}}^t(\theta, \eta_t^*) = \mathcal{J}_{\text{SM}}^t(\theta) - \frac{1}{2}\left\|\mathbb{E}_{q_t(x_t)}\left[s_\theta^t(x_t)\right]\right\|_2^2.
\tag{9}
$$

Since there is $\mathcal{J}_{\text{SM}}^t(\theta) = \mathcal{J}_{\text{DSM}}^t(\theta) + C^t$, we have $\mathcal{J}_{\text{DSM}}^t(\theta, \eta_t^*) = \mathcal{J}_{\text{DSM}}^t(\theta) - \frac{1}{2}\left\|\mathbb{E}_{q_t(x_t)}\left[s_\theta^t(x_t)\right]\right\|_2^2$ for the DSM objective. Similar calibration tricks are also valid under other model parametrizations and SM variants, as formally described in Appendix B.2.

**Remark.** For any pretrained score model $s_\theta^t(x_t)$, we can calibrate it into $s_\theta^t(x_t) - \mathbb{E}_{q_t(x_t)}[s_\theta^t(x_t)]$, which reduces the SM/DSM objectives at timestep $t$ by $\frac{1}{2}\left\|\mathbb{E}_{q_t(x_t)}[s_\theta^t(x_t)]\right\|_2^2$. The expectation of the calibrated score model is always zero, i.e., $\mathbb{E}_{q_t(x_t)}\left[s_\theta^t(x_t) - \mathbb{E}_{q_t(x_t)}[s_\theta^t(x_t)]\right] = 0$ holds for any $\theta$, which is consistent with $\mathbb{E}_{q_t(x_t)}[\nabla_{x_t} \log q_t(x_t)] = 0$ satisfied by data scores.

**Calibration preserves conservativeness.** A theoretical flaw of score-based modeling is that $s_\theta^t(x_t)$ may not correspond to a probability distribution. To solve this issue, Salimans and Ho [37] develop an energy-based model design, which utilizes the power of score-based modeling and simultaneously makes sure that $s_\theta^t(x_t)$ is conservative, i.e., there exists a probability distribution $p_\theta^t(x_t)$ such that $\forall x_t \in \mathbb{R}^k$, we have $s_\theta^t(x_t) = \nabla_{x_t} \log p_\theta^t(x_t)$. In this case, after we calibrate $s_\theta^t(x_t)$ by subtracting $\eta_t$, there is $s_\theta^t(x_t) - \eta_t = \nabla_{x_t} \log\left(\frac{p_\theta^t(x_t)}{\exp(x_t^\top \eta_t) Z_t(\theta)}\right)$, where $Z_t(\theta) = \int p_\theta^t(x_t) \exp\left(-x_t^\top \eta_t\right) dx_t$ represents the normalization factor. Intuitively, subtracting by $\eta_t$ corresponds to a shift in the vector space, so if $s_\theta^t(x_t)$ is conservative, its calibrated version $s_\theta^t(x_t) - \eta_t$ is also conservative.

**Conditional cases.** As to the conditional DPMs, we usually employ a conditional model $s_\theta^t(x_t, y)$, where $y \in \mathcal{Y}$ is the conditional context (e.g., class label or text prompt). To learn the conditional data score $\nabla_{x_t} \log q_t(x_t|y) = \nabla_{x_t} \log q_t(x_t, y)$, we minimize the SM objective defined as $\mathcal{J}_{\text{SM}}^t(\theta) \triangleq \frac{1}{2}\mathbb{E}_{q_t(x_t, y)}\left[\|s_\theta^t(x_t, y) - \nabla_{x_t} \log q_t(x_t, y)\|_2^2\right]$. Similar to the conclusion of $\mathbb{E}_{q_t(x_t)}[\nabla_{x_t} \log q_t(x_t)] = 0$, there is $\mathbb{E}_{q_t(x_t|y)}[\nabla_{x_t} \log q_t(x_t|y)] = 0$. To calibrate $s_\theta^t(x_t, y)$, we use the conditional term $\eta_t(y)$ and the calibrated SM objective is formulated as

$$
\begin{aligned}
\mathcal{J}_{\text{SM}}^t(\theta, \eta_t(y)) &\triangleq \frac{1}{2}\mathbb{E}_{q_t(x_t, y)}\left[\|s_\theta^t(x_t, y) - \eta_t(y) - \nabla_{x_t} \log q_t(x_t, y)\|_2^2\right] \\
&= \mathcal{J}_{\text{SM}}^t(\theta) - \mathbb{E}_{q_t(x_t, y)}\left[s_\theta^t(x_t, y)^\top \eta_t(y) + \frac{1}{2}\|\eta_t(y)\|_2^2\right],
\end{aligned}
\tag{10}
$$

and for any $y \in \mathcal{Y}$, the optimal $\eta_t^*(y)$ is given by $\eta_t^*(y) = \mathbb{E}_{q_t(x_t|y)}[\boldsymbol{s}_\theta^t(x_t, y)]$. We highlight the conditional context $y$ in contrast to the unconditional form in Eq. (7). After taking $\eta_t^*(y)$ into $\mathcal{J}_{\mathrm{SM}}^t(\theta, \eta_t(y))$, we have $\mathcal{J}_{\mathrm{SM}}^t(\theta, \eta_t^*(y)) = \mathcal{J}_{\mathrm{SM}}^t(\theta) - \frac{1}{2}\mathbb{E}_{q_t(y)}\left[\left\|\mathbb{E}_{q_t(x_t|y)}[\boldsymbol{s}_\theta^t(x_t, y)]\right\|_2^2\right]$. This conditional calibration form can naturally generalize to other model parametrizations and SM variants.

## 3.3 Likelihood of calibrated DPMs

Now we discuss the effects of calibration on model likelihood. Following the notations in Section 2.3, we use $p_0^{\mathrm{SDE}}(\theta, \eta_t)$ and $p_0^{\mathrm{ODE}}(\theta, \eta_t)$ to denote the distributions induced by the reverse-time SDE and ODE processes, respectively, where $\nabla_{x_t} \log q_t(x_t)$ is substituted with $\boldsymbol{s}_\theta^t(x_t) - \eta_t$.

**Likelihood of $p_0^{\mathrm{SDE}}(\theta, \eta_t)$.** Let $\mathcal{J}_{\mathrm{SM}}(\theta, \eta_t; g(t)^2) \triangleq \int_0^T g(t)^2 \mathcal{J}_{\mathrm{SM}}^t(\theta, \eta_t)dt$ be the total SM objective after the score model is calibrated by $\eta_t$, then according to Lemma 1, we have $\mathcal{D}_{\mathrm{KL}}\left(q_0 \| p_0^{\mathrm{SDE}}(\theta, \eta_t)\right) \leq \mathcal{J}_{\mathrm{SM}}(\theta, \eta_t; g(t)^2) + \mathcal{D}_{\mathrm{KL}}\left(q_T \| p_T\right)$. From the result in Eq. (9), there is

$$\mathcal{J}_{\mathrm{SM}}(\theta, \eta_t^*; g(t)^2) = \mathcal{J}_{\mathrm{SM}}(\theta; g(t)^2) - \frac{1}{2}\int_0^T g(t)^2 \left\|\mathbb{E}_{q_t(x_t)}[\boldsymbol{s}_\theta^t(x_t)]\right\|_2^2 dt. \tag{11}$$

Therefore, the likelihood $\mathcal{D}_{\mathrm{KL}}\left(q_0 \| p_0^{\mathrm{SDE}}(\theta, \eta_t^*)\right)$ after calibration has a lower upper bound of $\mathcal{J}_{\mathrm{SM}}(\theta, \eta_t^*; g(t)^2) + \mathcal{D}_{\mathrm{KL}}\left(q_T \| p_T\right)$, compared to the bound of $\mathcal{J}_{\mathrm{SM}}(\theta; g(t)^2) + \mathcal{D}_{\mathrm{KL}}\left(q_T \| p_T\right)$ for the original $\mathcal{D}_{\mathrm{KL}}\left(q_0 \| p_0^{\mathrm{SDE}}(\theta)\right)$. However, we need to clarify that $\mathcal{D}_{\mathrm{KL}}\left(q_0 \| p_0^{\mathrm{SDE}}(\theta, \eta_t^*)\right)$ may not necessarily smaller than $\mathcal{D}_{\mathrm{KL}}\left(q_0 \| p_0^{\mathrm{SDE}}(\theta)\right)$, since we can only compare their upper bounds.

**Likelihood of $p_0^{\mathrm{ODE}}(\theta, \eta_t)$.** Note that in Lemma 2, there is a term $\mathcal{J}_{\mathrm{Diff}}(\theta)$, which is usually small in practice since $\boldsymbol{s}_\theta^t(x_t)$ and $\nabla_{x_t} \log p_t^{\mathrm{ODE}}(x_t; \theta)$ are close. Thus, we have

$$\mathcal{D}_{\mathrm{KL}}\left(q_0 \| p_0^{\mathrm{ODE}}(\theta, \eta_t)\right) \approx \mathcal{J}_{\mathrm{SM}}(\theta, \eta_t; g(t)^2) + \mathcal{D}_{\mathrm{KL}}\left(q_T \| p_T\right),$$

and $\mathcal{D}_{\mathrm{KL}}\left(q_0 \| p_0^{\mathrm{ODE}}(\theta, \eta_t^*)\right)$ approximately achieves its lowest value. Lu et al. [28] show that $\mathcal{D}_{\mathrm{KL}}\left(q_0 \| p_0^{\mathrm{ODE}}(\theta)\right)$ can be further bounded by high-order SM objectives (as detailed in Appendix A.4), which depend on $\nabla_{x_t} \boldsymbol{s}_\theta^t(x_t)$ and $\nabla_{x_t} \mathbf{tr}\left(\nabla_{x_t} \boldsymbol{s}_\theta^t(x_t)\right)$. Since the calibration term $\eta_t$ is independent of $x_t$, i.e., $\nabla_{x_t} \eta_t = 0$, it does not affect the values of high-order SM objectives, and achieves a lower upper bound due to the lower value of the first-order SM objective.

## 3.4 Empirical learning fails to achieve $\mathbb{E}_{q_t(x_t)}[\boldsymbol{s}_\theta^t(x_t)] = 0$

A question that naturally arises is whether better architectures or learning algorithms for DPMs (e.g., EDMs [22]) could empirically achieve $\mathbb{E}_{q_t(x_t)}[\boldsymbol{s}_\theta^t(x_t)] = 0$ without calibration? The answer may be negative, since in practice we only have access to a *finite* dataset sampled from $q_0(x_0)$. More specifically, assuming that we have a training dataset $\mathbb{D} \triangleq \{x_0^n\}_{n=1}^N$ where $x_0^n \sim q_0(x_0)$, and defining the kernel density distribution induced by $\mathbb{D}$ as $q_t(x_t; \mathbb{D}) \propto \sum_{n=1}^N \mathcal{N}\left(\frac{x_t - \alpha_t x_0^n}{\sigma_t} | \mathbf{0}, \mathbf{I}\right)$. When the quantity of training data approaches infinity, we have $\lim_{N \to \infty} q_t(x_t; \mathbb{D}) = q_t(x_t)$ holds for $\forall t \in [0, T]$. Then the empirical DSM objective trained on $\mathbb{D}$ is written as

$$\mathcal{J}_{\mathrm{DSM}}^t(\theta; \mathbb{D}) \triangleq \frac{1}{2N}\sum_{n=1}^N \mathbb{E}_{q(\epsilon)}\left[\left\|\boldsymbol{s}_\theta^t(\alpha_t x_0^n + \sigma_t \epsilon) + \frac{\epsilon}{\sigma_t}\right\|_2^2\right], \tag{12}$$

and it is easy to show that the optimal solution for minimizing $\mathcal{J}_{\mathrm{DSM}}^t(\theta; \mathbb{D})$ satisfies (assuming $\boldsymbol{s}_\theta^t$ has universal model capacity) $\boldsymbol{s}_\theta^t(x_t) = \nabla_{x_t} \log q_t(x_t; \mathbb{D})$. Given a finite dataset $\mathbb{D}$, there is

$$\mathbb{E}_{q_t(x_t; \mathbb{D})}[\nabla_{x_t} \log q_t(x_t; \mathbb{D})] = 0, \text{ but typically } \mathbb{E}_{q_t(x_t)}[\nabla_{x_t} \log q_t(x_t; \mathbb{D})] \neq 0, \tag{13}$$

indicating that even if the score model is learned to be optimal, there is still $\mathbb{E}_{q_t(x_t)}[\boldsymbol{s}_\theta^t(x_t)] \neq 0$. Thus, the mis-calibration of DPMs is partially due to the *dataset bias*, i.e., during training we can only access a finite dataset $\mathbb{D}$ sampled from $q_0(x_0)$.

Furthermore, when trained on a finite dataset in practice, the learned model will not converge to the optimal solution [15], so there is typically $\boldsymbol{s}_\theta^t(x_t) \neq \nabla_{x_t} \log q_t(x_t; \mathbb{D})$ and $\mathbb{E}_{q_t(x_t; \mathbb{D})}[\boldsymbol{s}_\theta^t(x_t)] \neq 0$. After calibration, we can at least guarantee that $\mathbb{E}_{q_t(x_t; \mathbb{D})}\left[\boldsymbol{s}_\theta^t(x_t) - \mathbb{E}_{q_t(x_t; \mathbb{D})}[\boldsymbol{s}_\theta^t(x_t)]\right] = 0$ always holds on any finite dataset $\mathbb{D}$. In Figure 3, we demonstrate that even state-of-the-art EDMs still have non-zero and semantic $\mathbb{E}_{q_t(x_t)}[\boldsymbol{s}_\theta^t(x_t)]$, which emphasises the significance of calibrating DPMs.

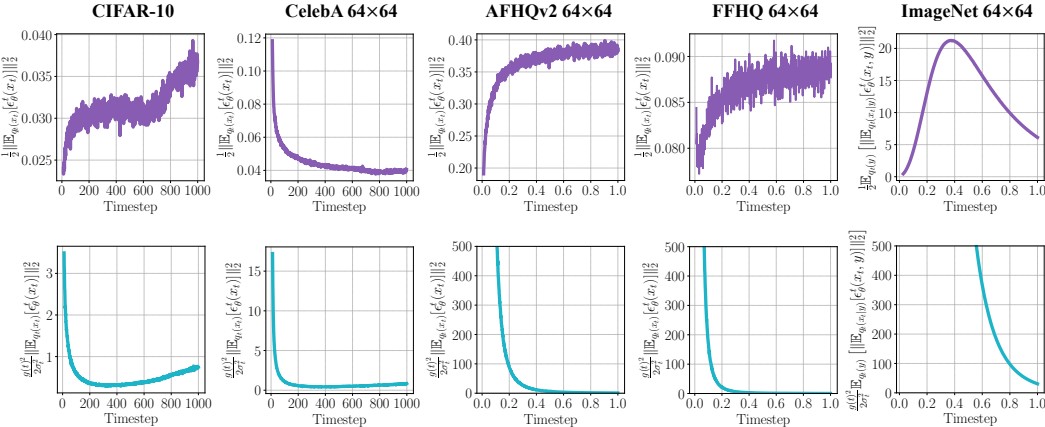

Figure 1: Time-dependent values of $\frac{1}{2}\|\mathbb{E}_{q_t(x_t)}\left[\boldsymbol{\epsilon}_\theta^t(x_t)\right]\|_2^2$ (the first row) and $\frac{g(t)^2}{2\sigma_t^2}\|\mathbb{E}_{q_t(x_t)}\left[\boldsymbol{\epsilon}_\theta^t(x_t)\right]\|_2^2$ (the second row) calculated on different datasets. The models on CIFAR-10 and CelebA is trained on discrete timesteps ($t = 0, 1, \cdots, 1000$), while those on AFHQv2, FFHQ, and ImageNet are trained on continuous timesteps ($t \in [0, 1]$). We convert data prediction $\boldsymbol{x}_\theta^t(x_t)$ into noise prediction $\boldsymbol{\epsilon}_\theta^t(x_t)$ based on $\boldsymbol{\epsilon}_\theta^t(x_t) = (x_t - \alpha_t \boldsymbol{x}_\theta^t(x_t))/\sigma_t$. The y-axis is clamped into $[0, 500]$.

## 3.5 Amortized computation of $\mathbb{E}_{q_t(x_t)}\left[\boldsymbol{s}_\theta^t(x_t)\right]$

By default, we are able to calculate and restore the value of $\mathbb{E}_{q_t(x_t)}\left[\boldsymbol{s}_\theta^t(x_t)\right]$ for a pretrained model $\boldsymbol{s}_\theta^t(x_t)$, where the selection of timestep $t$ is determined by the inference algorithm, and the expectation over $q_t(x_t)$ can be approximated by Monte Carlo sampling from a noisy training set. When we do not have access to training data, we can approximate the expectation using data generated from $p_t^{\text{ODE}}(x_t; \theta)$ or $p_t^{\text{SDE}}(x_t; \theta)$. Since we only need to calculate $\mathbb{E}_{q_t(x_t)}\left[\boldsymbol{s}_\theta^t(x_t)\right]$ once, the raised computational overhead is amortized as the number of generated samples increases.

**Dynamically recording.** In the preceding context, we focus primarily on post-training computing of $\mathbb{E}_{q_t(x_t)}\left[\boldsymbol{s}_\theta^t(x_t)\right]$. An alternative strategy would be to dynamically record $\mathbb{E}_{q_t(x_t)}\left[\boldsymbol{s}_\theta^t(x_t)\right]$ during the pretraining phase of $\boldsymbol{s}_\theta^t(x_t)$. Specifically, we could construct an auxiliary shallow network $h_\phi(t)$ parameterized by $\phi$, whose input is the timestep $t$. We define the expected mean squared error as

$$\mathcal{J}_{\text{Cal}}^t(\phi) \triangleq \mathbb{E}_{q_t(x_t)}\left[\|h_\phi(t) - \boldsymbol{s}_\theta^t(x_t)^\dagger\|_2^2\right], \tag{14}$$

where the superscript $\dagger$ denotes the stopping gradient and $\phi^*$ is the optimal solution of minimizing $\mathcal{J}_{\text{Cal}}^t(\phi)$ w.r.t. $\phi$, satisfying $h_{\phi^*}(t) = \eta_t^* = \mathbb{E}_{q_t(x_t)}\left[\boldsymbol{s}_\theta^t(x_t)\right]$ (assuming sufficient model capacity). The total training objective can therefore be expressed as $\mathcal{J}_{\text{SM}}(\theta; \lambda(t)) + \int_0^T \beta_t \cdot \mathcal{J}_{\text{Cal}}^t(\phi)$, where $\beta_t$ is a time-dependent trade-off coefficient for $t \in [0, T]$.

## 4 Experiments

In this section, we demonstrate that sample quality and model likelihood can be both improved by calibrating DPMs. Instead of establishing a new state-of-the-art, the purpose of our empirical studies is to testify the efficacy of our calibration technique as a simple way to repair DPMs.

### 4.1 Sample quality

**Setup.** We apply post-training calibration to discrete-time models trained on CIFAR-10 [25] and CelebA [27], which apply *parametrization of noise prediction* $\boldsymbol{\epsilon}_\theta^t(x_t)$. In the sampling phase, we employ DPM-Solver [29], an ODE-based sampler that achieves a promising balance between sample efficiency and image quality. Because our calibration directly acts on model scores, it is also compatible with other ODE/SDE-based samplers [3, 26], while we only focus on DPM-Solver cases in this paper. In accordance with the recommendation, we set the end time of DPM-Solver to $10^{-3}$ when the number of sampling steps is less than 15, and to $10^{-4}$ otherwise. Additional details can be

Table 1: Comparison on sample quality measured by FID ↓ with varying NFE on CIFAR-10. Experiments are conducted using a linear noise schedule on the discrete-time model from [17]. We consider three variants of DPM-Solver with different orders. The results with † mean the actual NFE is order $\times \lfloor \frac{\text{NFE}}{\text{order}} \rfloor$ which is smaller than the given NFE, following the setting in [29].

| Noise prediction | DPM-Solver | Number of evaluations (NFE) | | | | | | |
|---|---|---|---|---|---|---|---|---|
| | | 10 | 15 | 20 | 25 | 30 | 35 | 40 |
| $\epsilon_\theta^t(x_t)$ | 1-order | 20.49 | 12.47 | 9.72 | 7.89 | 6.84 | 6.22 | 5.75 |
| | 2-order | 7.35 | †4.52 | 4.14 | †3.92 | 3.74 | †3.71 | 3.68 |
| | 3-order | †23.96 | 4.61 | †3.89 | †3.73 | 3.65 | †3.65 | †3.60 |
| $\epsilon_\theta^t(x_t) - \mathbb{E}_{q_t(x_t)}\left[\epsilon_\theta^t(x_t)\right]$ | 1-order | 19.31 | 11.77 | 8.86 | 7.35 | 6.28 | 5.76 | 5.36 |
| | 2-order | **6.76** | †4.36 | 4.03 | †3.66 | 3.54 | †3.44 | 3.48 |
| | 3-order | †53.50 | **4.22** | †**3.32** | †**3.33** | 3.35 | †**3.32** | †**3.31** |

Table 2: Comparison on sample quality measured by FID ↓ with varying NFE on CelebA 64×64. Experiments are conducted using a linear noise schedule on the discrete-time model from [41]. The settings of DPM-Solver are the same as on CIFAR-10.

| Noise prediction | DPM-Solver | Number of evaluations (NFE) | | | | | | |
|---|---|---|---|---|---|---|---|---|
| | | 10 | 15 | 20 | 25 | 30 | 35 | 40 |
| $\epsilon_\theta^t(x_t)$ | 1-order | 16.74 | 11.85 | 7.93 | 6.67 | 5.90 | 5.38 | 5.01 |
| | 2-order | **4.32** | †3.98 | 2.94 | †2.88 | 2.88 | †2.88 | 2.84 |
| | 3-order | †11.92 | 3.91 | †2.84 | †2.76 | 2.82 | †2.81 | †2.85 |
| $\epsilon_\theta^t(x_t) - \mathbb{E}_{q_t(x_t)}\left[\epsilon_\theta^t(x_t)\right]$ | 1-order | 16.13 | 11.29 | 7.09 | 6.06 | 5.28 | 4.87 | 4.39 |
| | 2-order | 4.42 | †3.94 | 2.61 | †2.66 | 2.54 | †2.52 | **2.49** |
| | 3-order | †35.47 | **3.62** | †**2.33** | †**2.43** | 2.40 | †**2.43** | †**2.49** |

found in Lu et al. [29]. By default, we employ the FID score [16] to quantify the sample quality using 50,000 samples. Typically, a lower FID indicates a higher sample quality. In addition, in Table 3, we evaluate using other metrics such as sFID [30], IS [39], and Precision/Recall [36].

**Computing** $\mathbb{E}_{q_t(x_t)}\left[\epsilon_\theta^t(x_t)\right]$**.** To estimate the expectation over $q_t(x_t)$, we construct $x_t = \alpha_t x_0 + \sigma_t \epsilon$, where $x_0 \sim q_0(x_0)$ is sampled from the training set and $\epsilon \sim \mathcal{N}(\epsilon|\mathbf{0}, \mathbf{I})$ is sampled from a standard Gaussian distribution. The selection of timestep $t$ depends on the sampling schedule of DPM-Solver. The computed values of $\mathbb{E}_{q_t(x_t)}\left[\epsilon_\theta^t(x_t)\right]$ are restored in a dictionary and warped into the output layers of DPMs, allowing existing inference pipelines to be reused.

We first calibrate the model trained by Ho et al. [17] on the CIFAR-10 dataset and compare it to the original one for sampling with DPM-Solvers. We conduct a systematical study with varying NFE (i.e., number of function evaluations) and solver order. The results are presented in Tables 1 and 3. After calibrating the model, the sample quality is consistently enhanced, which demonstrates the significance of doing so and the efficacy of our method. We highlight the significant improvement in sample quality (4.61→4.22 when using 15 NFE and 3-order DPM-Solver; 3.89→3.32 when using 20 NFE and 3-order DPM-Solver). After model calibration, the number of steps required to achieve convergence for a 3-order DPM-Solver is reduced from ≥30 to 20, making our method a new option for expediting the sampling of DPMs. In addition, as a point of comparison, the 3-order DPM-Solver with 1,000 NFE can only yield an FID score of 3.45 when using the original model, which, along with the results in Table 1, indicates that model calibration helps to improve the convergence of sampling.

Then, we conduct experiments with the discrete-time model trained on the CelebA 64x64 dataset by Song et al. [41]. The corresponding sample quality comparison is shown in Table 2. Clearly, model calibration brings significant gains (3.91→3.62 when using 15 NFE and 3-order DPM-Solver; 2.84→2.33 when using 20 NFE and 3-order DPM-Solver) that are consistent with those on the CIFAR-10 dataset. This demonstrates the prevalence of the mis-calibration issue in existing DPMs and the efficacy of our correction. We still observe that model calibration improves convergence of sampling, and as shown in Figure 2, our calibration could help to reduce ambiguous generations. More generated images are displayed in Appendix C.

Table 3: Comparison on sample quality measured by different metrics, including FID ↓, sFID ↓, inception score (IS) ↑, precision ↑ and recall ↑ with varying NFE on CIFAR-10. We use Base to denote the baseline $\epsilon_\theta^t(x_t)$ and Ours to denote calibrated score $\epsilon_\theta^t(x_t) - \mathbb{E}_{q_t(x_t)}[\epsilon_\theta^t(x_t)]$. The sampler is DPM-Solver with different orders. Note that FID is computed by the PyTorch checkpoint of Inception-v3, while sFID/IS/Precision/Recall are computed by the Tensorflow checkpoint of Inception-v3 following *github.com/kynkaat/improved-precision-and-recall-metric*.

| Method | | Number of evaluations (NFE) | | | | | | | | | | | | | | |
| | | 20 | | | | | 25 | | | | | 30 | | | | |
| | | FID | sFID | IS | Pre. | Rec. | FID | sFID | IS | Pre. | Rec. | FID | sFID | IS | Pre. | Rec. |
| | 1-ord. | 9.72 | 6.03 | 8.49 | 0.641 | 0.542 | 7.89 | 5.45 | 8.68 | 0.644 | 0.556 | 6.84 | 5.12 | 8.76 | 0.650 | 0.565 |
| Base | 2-ord. | 4.14 | 4.36 | 9.15 | 0.654 | 0.590 | 3.92 | 4.22 | 9.17 | 0.657 | 0.591 | 3.74 | 4.18 | 9.20 | 0.658 | 0.591 |
| | 3-ord. | 3.89 | 4.18 | 9.29 | 0.652 | 0.597 | 3.73 | 4.15 | 9.21 | 0.657 | 0.595 | 3.65 | 4.12 | 9.22 | 0.658 | 0.593 |
| | 1-ord. | 8.86 | 6.01 | 8.56 | 0.649 | 0.544 | 7.35 | 5.42 | 8.76 | 0.653 | 0.560 | 6.28 | 5.09 | 8.84 | 0.653 | 0.568 |
| Ours | 2-ord. | 4.03 | 4.31 | 9.17 | **0.661** | 0.592 | 3.66 | 4.20 | 9.20 | 0.664 | 0.594 | 3.54 | 4.14 | 9.23 | **0.662** | 0.599 |
| | 3-ord. | **3.32** | 4.14 | 9.38 | 0.657 | **0.603** | 3.33 | 4.11 | 9.28 | 0.665 | 0.597 | 3.35 | 4.08 | 9.27 | 0.662 | **0.600** |

w/o calibration (**baseline**)

- - - - - - - - - - - - - - - - - - - - - - - - - - - - - - - - - - - - - - - - - - - - -

w/ calibration (**ours**)

Figure 2: Selected images on CIFAR-10 (generated with NFE = 20 using 3-order DPM-Solver) demonstrating that our calibration could reduce ambiguous generations, such as generations that resemble both horse and dog. However, we must emphasize that not all generated images have a visually discernible difference before and after calibration.

## 4.2  Model likelihood

As described in Section 3.3, calibration contributes to reducing the SM objective, thereby decreasing the upper bound of the KL divergence between model distribution at timestep $t = 0$ (either $p_0^{\text{SDE}}(\theta, \eta_t^*)$ or $p_0^{\text{ODE}}(\theta, \eta_t^*)$) and data distribution $q_0$. Consequently, it aids in raising the lower bound of model likelihood. In this subsection, we examine such effects by evaluating the aforementioned DPMs on the CIFAR-10 and CelebA datasets. We also conduct experiments with continuous-time models trained by Karras et al. [22] on AFHQv2 64×64 [7], FFHQ 64×64 [21], and ImageNet 64×64 [9] datasets considering their top performance. These models apply parametrization of data prediction $x_\theta^t(x_t)$, and for consistency, we convert it to align with $\epsilon_\theta^t(x_t)$ based on the relationship $\epsilon_\theta^t(x_t) = (x_t - \alpha_t x_\theta^t(x_t))/\sigma_t$, as detailed in Kingma et al. [23] and Appendix B.2.

Given that we employ noise prediction models in practice, we first estimate $\frac{1}{2}\|\mathbb{E}_{q_t(x_t)}[\epsilon_\theta^t(x_t)]\|_2^2$ at timestep $t \in [0, T]$, which reflects the decrement on the SM objective at $t$ according to Eq. (9) (up to a scaling factor of $1/\sigma_t^2$). We approximate the expectation using Monte Carlo (MC) estimation with training data points. The results are displayed in the first row of Figure 1. Notably, the value of $\frac{1}{2}\|\mathbb{E}_{q_t(x_t)}[\epsilon_\theta^t(x_t)]\|_2^2$ varies significantly along with timestep $t$: it decreases relative to $t$ for CelebA but increases in all other cases (except for $t \in [0.4, 1.0]$ on ImageNet 64×64). Ideally, there should be $\frac{1}{2}\|\mathbb{E}_{q_t(x_t)}[\nabla_{x_t} \log q_t(x_t)]\|_2^2 = 0$ at any $t$. Such inconsistency reveals that mis-calibration issues exist in general, although the phenomenon may vary across datasets and training mechanisms.

Then, we quantify the gain of model calibration on increasing the lower bound of model likelihood, which is $\frac{1}{2}\int_0^T g(t)^2 \|\mathbb{E}_{q_t(x_t)}[s_\theta^t(x_t)]\|_2^2 dt$ according to Eq. (11). We first rewrite it with the model parametrization of noise prediction $\epsilon_\theta^t(x_t)$, and it can be straightforwardly demonstrated that it equals $\int_0^T \frac{g(t)^2}{2\sigma_t^2}\|\mathbb{E}_{q_t(x_t)}[\epsilon_\theta^t(x_t)]\|_2^2$. Therefore, we calculate the value of $\frac{g(t)^2}{2\sigma_t^2}\|\mathbb{E}_{q_t(x_t)}[\epsilon_\theta^t(x_t)]\|_2^2$ using MC

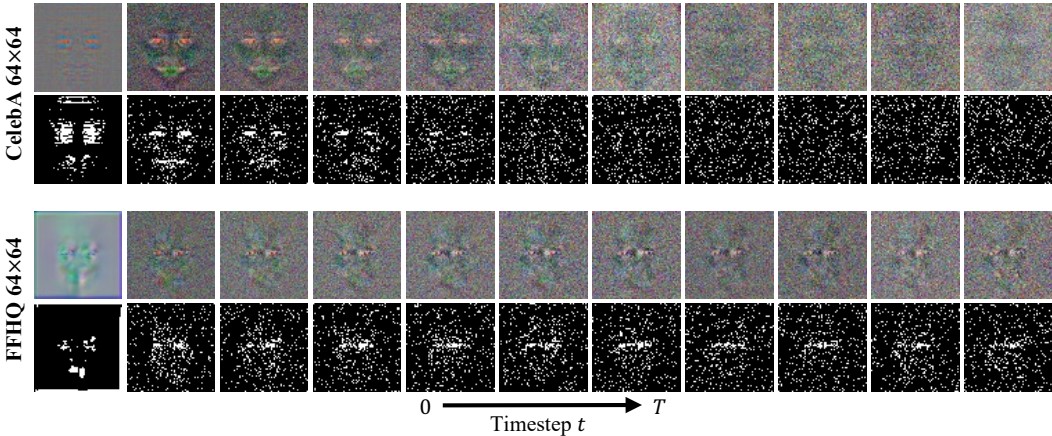

Figure 3: Visualization of the expected predicted noises with increasing $t$. For each dataset, the first row displays $\mathbb{E}_{q_t(x_t)}[\epsilon_\theta^t(x_t)]$ (after normalization) and the second row highlights the top-10% pixels that $\mathbb{E}_{q_t(x_t)}[\epsilon_\theta^t(x_t)]$ has high values. The DPM on CelebA is a discrete-time model with 1000 timesteps [41] and that on FFHQ is a continuous-time one [22].

estimation and report the results in the second row of Figure 1. The integral is represented by the area under the curve (i.e., the gain of model calibration on the lower bound of model likelihood). Various datasets and model architectures exhibit non-trivial gains, as observed. In addition, we notice that the DPMs trained by Karras et al. [22] show patterns distinct from those of DDPM [17] and DDIM [41], indicating that different DPM training mechanisms may result in different mis-calibration effects.

**Visualizing $\mathbb{E}_{q_t(x_t)}[\epsilon_\theta^t(x_t)]$.** To better understand the inductive bias learned by DPMs, we visualize the expected predicted noises $\mathbb{E}_{q_t(x_t)}[\epsilon_\theta^t(x_t)]$ for timestep from 0 to $T$, as seen in Figure 3. For each dataset, the first row normalizes the values of $\mathbb{E}_{q_t(x_t)}[\epsilon_\theta^t(x_t)]$ into $[0, 255]$; the second row calculates pixel-wise norm (across RGB channels) and highlights the top-10% locations with the highest norm. As we can observe, on facial datasets like CelebA and FFHQ, there are obvious facial patterns inside $\mathbb{E}_{q_t(x_t)}[\epsilon_\theta^t(x_t)]$, while on other datasets like CIFAR-10, ImageNet, as well as the animal face dataset AFHQv2, the patterns inside $\mathbb{E}_{q_t(x_t)}[\epsilon_\theta^t(x_t)]$ are more like random noises. Besides, the facial patterns in Figure 3 are more significant when $t$ is smaller, and become blurry when $t$ is close to $T$. This phenomenon may be attributed to the bias of finite training data, which is detrimental to generalization during sampling and justifies the importance of calibration as described in Section 3.4.

### 4.3 Ablation studies

We conduct ablation studies focusing on the estimation methods of $\mathbb{E}_{q_t(x_t)}[\epsilon_\theta^t(x_t)]$.

**Estimating $\mathbb{E}_{q_t(x_t)}[\epsilon_\theta^t(x_t)]$ with partial training data.** In the post-training calibration setting, our primary algorithmic change is to subtract the calibration term $\mathbb{E}_{q_t(x_t)}[\epsilon_\theta^t(x_t)]$ from the pretrained DPMs' output. In the aforementioned studies, the expectation in $\mathbb{E}_{q_t(x_t)}[\epsilon_\theta^t(x_t)]$ (or its variant of other model parametrizations) is approximated with MC estimation using all training images. However, there may be situations where training data are (partially) inaccessible. To evaluate the effectiveness of our method under these cases, we examine the number of training images used to estimate the calibration term on CIFAR-10. To determine the quality of the estimated calibration term, we sample from the calibrated models using a 3-order DPM-Solver running for 20 steps and evaluate the corresponding FID score. The results are listed in the left part of Table 4. As observed, we need to use the majority of training images (at least $\geq 20,000$) to estimate the calibration term. We deduce that this is because the CIFAR-10 images are rich in diversity, necessitating a non-trivial number of training images to cover the various modes and produce a nearly unbiased calibration term.

**Estimating $\mathbb{E}_{q_t(x_t)}[\epsilon_\theta^t(x_t)]$ with generated data.** In the most extreme case where we do not have access to any training data (e.g., due to privacy concerns), we could still estimate the expectation over $q_t(x_t)$ with data generated from $p_0^{\text{ODE}}(x_0; \theta)$ or $p_0^{\text{SDE}}(x_0; \theta)$. Specifically, under the hypothesis that $p_0^{\text{ODE}}(x_0; \theta) \approx q_0(x_0)$ (DPM-Solver is an ODE-based sampler), we first generate $\widetilde{x}_0 \sim p_0^{\text{ODE}}(x_0; \theta)$

Table 4: Sample quality varies w.r.t. the number of training images (left part) and generated images (right part) used to estimate the calibration term on CIFAR-10. In the generated data case, the images used to estimate the calibration term $\mathbb{E}_{q_t(x_t)}[\epsilon_\theta^t(x_t)]$ is crafted with 50 sampling steps by a 3-order DPM-Solver.

| Training data | | Generated data | |
|---|---|---|---|
| # of samples | FID ↓ | # of samples | FID ↓ |
| 500 | 55.38 | 2,000 | 8.80 |
| 1,000 | 18.72 | 5,000 | 4.53 |
| 2,000 | 8.05 | 10,000 | 3.78 |
| 5,000 | 4.31 | 20,000 | **3.31** |
| 10,000 | 3.47 | 50,000 | 3.46 |
| 20,000 | **3.25** | 100,000 | 3.47 |
| 50,000 | 3.32 | 200,000 | 3.46 |

Figure 4: Dynamically recording $\mathbb{E}_{q_t(x_t)}[\epsilon_\theta^t(x_t)]$. During training, the mean square error between the ground truth and the outputs of a shallow network for recording the calibration terms rapidly decreases, across different timesteps $t$.

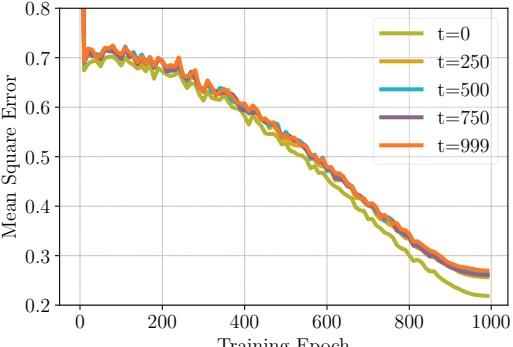

and construct $\widetilde{x}_t = \alpha_t \widetilde{x}_0 + \sigma_t \epsilon$, where $\widetilde{x}_t \sim p_t^{\text{ODE}}(x_t; \theta)$. Then, the expectation over $q_t(x_t)$ could be approximated by the expectation over $p_t^{\text{ODE}}(x_t; \theta)$.

Empirically, on the CIFAR-10 dataset, we adopt a 3-order DPM-Solver to generate a set of samples from the pretrained model of Ho et al. [17], using a relatively large number of sampling steps (e.g., 50 steps). This set of generated data is used to calculate the calibration term $\mathbb{E}_{q_t(x_t)}[\epsilon_\theta^t(x_t)]$. Then, we obtain the calibrated model $\epsilon_\theta^t(x_t) - \mathbb{E}_{q_t(x_t)}[\epsilon_\theta^t(x_t)]$ and craft new images based on a 3-order 20-step DPM-Solver. In the right part of Table 4, we present the results of an empirical investigation into how the number of generated images influences the quality of model calibration.

Using the same sampling setting, we also provide two reference points: 1) the originally mis-calibrated model can reach the FID score of 3.89, and 2) the model calibrated with training data can reach the FID score of 3.32. Comparing these results reveals that the DPM calibrated with a large number of high-quality generations can achieve comparable FID scores to those calibrated with training samples (see the result of using 20,000 generated images). Additionally, it appears that using more generations is not advantageous. This may be because the generations from DPMs, despite being known to cover diverse modes, still exhibit semantic redundancy and deviate slightly from the data distribution.

**Dynamical recording.** We simulate the proposed dynamical recording technique. Specifically, we use a 3-layer MLP of width 512 to parameterize the aforementioned network $h_\phi(t)$ and train it with an Adam optimizer [24] to approximate the expected predicted noises $\mathbb{E}_{q_t(x_t)}[\epsilon_\theta^t(x_t)]$, where $\epsilon_\theta^t(x_t)$ comes from the pretrained noise prediction model on CIFAR-10 [17]. The training of $h_\phi(t)$ runs for 1,000 epochs. Meanwhile, using the training data, we compute the expected predicted noises with MC estimation and treat them as the ground truth. In Figure 4, we compare them to the outputs of $h_\phi(t)$ and visualize the disparity measured by mean square error. As demonstrated, as the number of training epochs increases, the network $h_\phi(t)$ quickly converges and can form a relatively reliable approximation to the ground truth. Dynamic recording has a distinct advantage of being able to be performed during the training of DPMs to enable immediate generation. We clarify that better timestep embedding techniques and NN architectures can improve approximation quality even further.

## 5 Discussion

We propose a straightforward method for calibrating any pretrained DPM that can provably reduce the values of SM objectives and, as a result, induce higher values of lower bounds for model likelihood. We demonstrate that the mis-calibration of DPMs may be inherent due to the dataset bias and/or sub-optimally learned model scores. Our findings also provide a potentially new metric for assessing a diffusion model by its degree of "uncalibration", namely, how far the learned scores deviate from the essential properties (e.g., the expected data scores should be zero).

**Limitations.** While our calibration method provably improves the model's likelihood, it does not necessarily yield a lower FID score, as previously discussed [45]. Besides, for text-to-image generation, post-training computation of $\mathbb{E}_{q_t(x_t|y)}[s_\theta^t(x_t, y)]$ becomes infeasible due to the exponentially large number of conditions $y$, necessitating dynamic recording with multimodal modules.

## Acknowledgements

Zhijie Deng was supported by Natural Science Foundation of Shanghai (No. 23ZR1428700) and the Key Research and Development Program of Shandong Province, China (No. 2023CXGC010112).

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

# A Detailed derivations

In this section, we provide detailed derivations for the Theorem and equations shown in the main text. We follow the regularization assumptions listed in Song et al. [45].

## A.1 Proof of Theorem 1

*Proof.* For any two timesteps $0 \le s < t \le T$, i.e., the transition probability from $x_s$ to $x_t$ is written as $q_{st}(x_t|x_s) = \mathcal{N}\left(x_t \middle| \alpha_{t|s} x_s, \sigma_{t|s}^2 \mathbf{I}\right)$, where $\alpha_{t|s} = \frac{\alpha_t}{\alpha_s}$ and $\sigma_{t|s}^2 = \sigma_t^2 - \alpha_{t|s}^2 \sigma_s^2$. The marginal distribution $q_t(x_t) = \int q_{st}(x_t|x_s) q_s(x_s) dx_s$ and we have

$$
\begin{aligned}
\nabla_{x_t} \log q_t(x_t) &= \frac{1}{\alpha_{t|s}} \nabla_{\alpha_{t|s}^{-1} x_t} \log \left( \frac{1}{\alpha_{t|s}^k} \mathbb{E}_{\mathcal{N}\left(x_s \middle| \alpha_{t|s}^{-1} x_t, \alpha_{t|s}^{-2} \sigma_{t|s}^2 \mathbf{I}\right)} [q_s(x_s)] \right) \\
&= \frac{1}{\alpha_{t|s}} \nabla_{\alpha_{t|s}^{-1} x_t} \log \left( \mathbb{E}_{\mathcal{N}\left(\eta \middle| 0, \alpha_{t|s}^{-2} \sigma_{t|s}^2 \mathbf{I}\right)} \left[ q_s(\alpha_{t|s}^{-1} x_t + \eta) \right] \right) \\
&= \frac{\mathbb{E}_{\mathcal{N}\left(\eta \middle| 0, \alpha_{t|s}^{-2} \sigma_{t|s}^2 \mathbf{I}\right)} \left[ \nabla_{\alpha_{t|s}^{-1} x_t} q_s(\alpha_{t|s}^{-1} x_t + \eta) \right]}{\alpha_{t|s} \mathbb{E}_{\mathcal{N}\left(\eta \middle| 0, \alpha_{t|s}^{-2} \sigma_{t|s}^2 \mathbf{I}\right)} \left[ q_s(\alpha_{t|s}^{-1} x_t + \eta) \right]} \\
&= \frac{\mathbb{E}_{\mathcal{N}\left(\eta \middle| 0, \alpha_{t|s}^{-2} \sigma_{t|s}^2 \mathbf{I}\right)} \left[ q_s(\alpha_{t|s}^{-1} x_t + \eta) \nabla_{\alpha_{t|s}^{-1} x_t + \eta} \log q_s(\alpha_{t|s}^{-1} x_t + \eta) \right]}{\alpha_{t|s} \mathbb{E}_{\mathcal{N}\left(\eta \middle| 0, \alpha_{t|s}^{-2} \sigma_{t|s}^2 \mathbf{I}\right)} \left[ q_s(\alpha_{t|s}^{-1} x_t + \eta) \right]} \\
&= \frac{\mathbb{E}_{\mathcal{N}\left(x_s \middle| \alpha_{t|s}^{-1} x_t, \alpha_{t|s}^{-2} \sigma_{t|s}^2 \mathbf{I}\right)} [q_s(x_s) \nabla_{x_s} \log q_s(x_s)]}{\alpha_{t|s} \mathbb{E}_{\mathcal{N}\left(x_s \middle| \alpha_{t|s}^{-1} x_t, \alpha_{t|s}^{-2} \sigma_{t|s}^2 \mathbf{I}\right)} [q_s(x_s)]} \\
&= \frac{\int \mathcal{N}\left(x_t \middle| \alpha_{t|s} x_s, \sigma_{t|s}^2 \mathbf{I}\right) q_s(x_s) \nabla_{x_s} \log q_s(x_s) dx_s}{\alpha_{t|s} \int \mathcal{N}\left(x_t \middle| \alpha_{t|s} x_s, \sigma_{t|s}^2 \mathbf{I}\right) q_s(x_s) dx_s} \\
&= \frac{1}{\alpha_{t|s}} \mathbb{E}_{q_{st}(x_s|x_t)} [\nabla_{x_s} \log q_s(x_s)] .
\end{aligned}
\tag{15}
$$

Note that when the transition probability $q_{st}(x_t|x_s)$ corresponds to a well-defined forward process, there is $\alpha_t > 0$ for $\forall t \in [0, T]$, and thus we achieve $\alpha_t \nabla_{x_t} \log q_t(x_t) = \mathbb{E}_{q_{st}(x_s|x_t)} [\alpha_s \nabla_{x_s} \log q_s(x_s)]$. □

## A.2 Proof of $\mathbb{E}_{q_0(x_0)} [\nabla_{x_0} \log q_0(x_0)] = 0$

*Proof.* The input variable $x \in \mathbb{R}^k$ and $q_0(x_0) \in \mathcal{C}^2$, where $\mathcal{C}^2$ denotes the family of functions with continuous second-order derivatives.[1] We use $x^i$ denote the $i$-th element of $x$, then we can derive the expectation

$$
\begin{aligned}
\mathbb{E}_{q_0(x_0)} \left[ \frac{\partial}{\partial x_0^i} \log q_0(x_0) \right] &= \int \cdots \int q_0(x_0) \frac{\partial}{\partial x_0^i} \log q_0(x_0) dx_0^1 dx_0^2 \cdots dx_0^k \\
&= \int \cdots \int \frac{\partial}{\partial x_0^i} q_0(x_0) dx_0^1 dx_0^2 \cdots dx_0^k \\
&= \int \frac{\partial}{\partial x_0^i} \left( \int q_0(x_0^i, x_0^{\backslash i}) dx_0^{\backslash i} \right) dx_0^i \\
&= \int \frac{d}{dx_0^i} q_0(x_0^i) dx_0^i = 0,
\end{aligned}
\tag{16}
$$

---

[1]This continuously differentiable assumption can be satisfied by adding a small Gaussian noise (e.g., with variance of 0.0001) on the original data distribution, as done in Song and Ermon [42].

where $x_0^{\backslash i}$ denotes all the $k-1$ elements in $x_0$ except for the $i$-th one. The last equation holds under the boundary condition that $\lim_{x_0^i \to \infty} q_0(x_0^i) = 0$ hold for any $i \in [K]$. Thus, we achieve the conclusion that $\mathbb{E}_{q_0(x_0)} [\nabla_{x_0} \log q_0(x_0)] = 0$. $\qquad\square$

### A.3 Concentration bounds

We describe concentration bounds [11, 1] of the martingale $\alpha_t \nabla_{x_t} \log q_t(x_t)$.

**Azuma's inequality.** For discrete reverse timestep $t = T, T-1, \cdots, 0$, Assuming that there exist constants $0 < c_1, c_2, \cdots, < \infty$ such that for the $i$-th element of $x$,

$$A_t \leq \frac{\partial}{\partial x_{t-1}^i} \alpha_{t-1} \log q_{t-1}(x_{t-1}) - \frac{\partial}{\partial x_t^i} \alpha_t \log q_t(x_t) \leq B_t \text{ and } B_t - A_t \leq c_t \tag{17}$$

almost surely. Then $\forall \epsilon > 0$, the probability (note that $\alpha_0 = 1$)

$$P\left( \left| \frac{\partial}{\partial x_0^i} \log q_0(x_0) - \frac{\partial}{\partial x_T^i} \alpha_T \log q_T(x_T) \right| \geq \epsilon \right) \leq 2 \exp\left( -\frac{2\epsilon^2}{\sum_{t=1}^T c_t^2} \right). \tag{18}$$

Specially, considering that $q_T(x_T) \approx \mathcal{N}(x_T|0, \widetilde{\sigma}^2 \mathbf{I})$, there is $\frac{\partial}{\partial x_T^i} \log q_T(x_T) \approx -\frac{x_T^i}{\widetilde{\sigma}^2}$. Thus, we can approximately obtain

$$P\left( \left| \frac{\partial}{\partial x_0^i} \log q_0(x_0) + \frac{\alpha_T x_T^i}{\widetilde{\sigma}^2} \right| \geq \epsilon \right) \leq 2 \exp\left( -\frac{2\epsilon^2}{\sum_{t=1}^T c_t^2} \right). \tag{19}$$

**Doob's inequality.** For continuous reverse timestep $t$ from $T$ to $0$, if the sample paths of the martingale are almost surely right-continuous, then for the $i$-th element of $x$ we have (note that $\alpha_0 = 1$)

$$P\left( \sup_{0 \leq t \leq T} \frac{\partial}{\partial x_t^i} \alpha_t \log q_t(x_t) \geq C \right) \leq \frac{\mathbb{E}_{q_0(x_0)} \left[ \max\left( \frac{\partial}{\partial x_0^i} \log q_0(x_0), 0 \right) \right]}{C}. \tag{20}$$

### A.4 High-order SM objectives

Lu et al. [28] show that the KL divergence $\mathcal{D}_{\mathrm{KL}} \left( q_0 \| p_0^{\mathrm{ODE}}(\theta) \right)$ can be bounded as

$$\mathcal{D}_{\mathrm{KL}} \left( q_0 \| p_0^{\mathrm{ODE}}(\theta) \right) \leq \mathcal{D}_{\mathrm{KL}} \left( q_T \| p_T \right) + \sqrt{\mathcal{J}_{\mathrm{SM}}(\theta; g(t)^2)} \cdot \sqrt{\mathcal{J}_{\mathrm{Fisher}}(\theta)}, \tag{21}$$

where $\mathcal{J}_{\mathrm{Fisher}}(\theta)$ is a weighted sum of Fisher divergence between $q_t(x_t)$ and $p_t^{\mathrm{ODE}}(\theta)$ as

$$\mathcal{J}_{\mathrm{Fisher}}(\theta) = \frac{1}{2} \int_0^T g(t)^2 D_F \left( q_t \| p_t^{\mathrm{ODE}}(\theta) \right) dt. \tag{22}$$

Moreover, Lu et al. [28] prove that if $\forall t \in [0, T]$ and $\forall x_t \in \mathbb{R}^k$, there exist a constant $C_F$ such that the spectral norm of Hessian matrix $\| \nabla_{x_t}^2 \log p_t^{\mathrm{ODE}}(x_t; \theta) \|_2 \leq C_F$, and there exist $\delta_1, \delta_2, \delta_3 > 0$ such that

$$\begin{aligned}
\| s_\theta^t(x_t) - \nabla_{x_t} \log q_t(x_t) \|_2 &\leq \delta_1, \\
\| \nabla_{x_t} s_\theta^t(x_t) - \nabla_{x_t}^2 \log q_t(x_t) \|_F &\leq \delta_2, \\
\| \nabla_{x_t} \mathbf{tr} \left( \nabla_{x_t} s_\theta^t(x_t) \right) - \nabla_{x_t} \mathbf{tr} \left( \nabla_{x_t}^2 \log q_t(x_t) \right) \|_2 &\leq \delta_3,
\end{aligned} \tag{23}$$

where $\| \cdot \|_F$ is the Frobenius norm of matrix. Then there exist a function $U(t; \delta_1, \delta_2, \delta_3, q)$ that independent of $\theta$ and strictly increasing (if $g(t) \neq 0$) w.r.t. $\delta_1, \delta_2$, and $\delta_3$, respectively, such that the Fisher divergence can be bounded as $D_F \left( q_t \| p_t^{\mathrm{ODE}}(\theta) \right) \leq U(t; \delta_1, \delta_2, \delta_3, q)$.

**The case after calibration.** When we impose the calibration term $\eta_t^* = \mathbb{E}_{q_t(x_t)} [s_\theta^t(x_t)]$ to get the score model $s_\theta^t(x_t) - \eta_t^*$, there is $\nabla_{x_t} \eta_t^* = 0$ and thus $\nabla_{x_t} (s_\theta^t(x_t) - \eta_t^*) = \nabla_{x_t} s_\theta^t(x_t)$. Then we have

$$\begin{aligned}
\| s_\theta^t(x_t) - \eta_t^* - \nabla_{x_t} \log q_t(x_t) \|_2 &\leq \delta_1' \leq \delta_1, \\
\| \nabla_{x_t} \left( s_\theta^t(x_t) - \eta_t^* \right) - \nabla_{x_t}^2 \log q_t(x_t) \|_F &\leq \delta_2, \\
\| \nabla_{x_t} \mathbf{tr} \left( \nabla_{x_t} \left( s_\theta^t(x_t) - \eta_t^* \right) \right) - \nabla_{x_t} \mathbf{tr} \left( \nabla_{x_t}^2 \log q_t(x_t) \right) \|_2 &\leq \delta_3.
\end{aligned} \tag{24}$$

From these, we know that the Fisher divergence $D_F\left(q_t\|p_t^{\text{ODE}}(\theta, \eta_t^*)\right) \leq U(t; \delta_1', \delta_2, \delta_3, q) \leq U(t; \delta_1, \delta_2, \delta_3, q)$, namely, $D_F\left(q_t\|p_t^{\text{ODE}}(\theta, \eta_t^*)\right)$ has a lower upper bound compared to $D_F\left(q_t\|p_t^{\text{ODE}}(\theta)\right)$. Consequently, we can get lower upper bounds for both $\mathcal{J}_{\text{Fisher}}(\theta, \eta_t^*)$ and $\mathcal{D}_{\text{KL}}\left(q_0\|p_0^{\text{ODE}}(\theta, \eta_t^*)\right)$, compared to $\mathcal{J}_{\text{Fisher}}(\theta)$ and $\mathcal{D}_{\text{KL}}\left(q_0\|p_0^{\text{ODE}}(\theta)\right)$, respectively.

## B  Model parametrization

This section introduces different parametrizations used in diffusion models and provides their calibrated instantiations.

### B.1  Preliminary

Along the research routine of diffusion models, different model parametrizations have been used, including score prediction $\boldsymbol{s}_\theta^t(x_t)$ [42, 46], noise prediction $\boldsymbol{\epsilon}_\theta^t(x_t)$ [17, 33], data prediction $\boldsymbol{x}_\theta^t(x_t)$ [23, 32], and velocity prediction $\boldsymbol{v}_\theta^t(x_t)$ [38, 18]. Taking the DSM objective as the training loss, its instantiation at timestep $t \in [0, T]$ is written as

$$\mathcal{J}_{\text{DSM}}^t(\theta) = \begin{cases} \frac{1}{2}\mathbb{E}_{q_0(x_0), q(\epsilon)}\left[\|\boldsymbol{s}_\theta^t(x_t) + \frac{\epsilon}{\sigma_t}\|_2^2\right], & \text{score prediction;} \\ \frac{\alpha_t^2}{2\sigma_t^4}\mathbb{E}_{q_0(x_0), q(\epsilon)}\left[\|\boldsymbol{x}_\theta^t(x_t) - x_0\|_2^2\right], & \text{data prediction;} \\ \frac{1}{2\sigma_t^2}\mathbb{E}_{q_0(x_0), q(\epsilon)}\left[\|\boldsymbol{\epsilon}_\theta^t(x_t) - \epsilon\|_2^2\right], & \text{noise prediction;} \\ \frac{\alpha_t^2}{2\sigma_t^2}\mathbb{E}_{q_0(x_0), q(\epsilon)}\left[\|\boldsymbol{v}_\theta^t(x_t) - (\alpha_t\epsilon - \sigma_t x_0)\|_2^2\right], & \text{velocity prediction.} \end{cases} \quad (25)$$

### B.2  Calibrated instantiation

Under different model parametrizations, we can derive the optimal calibration terms $\eta_t^*$ that minimizing $\mathcal{J}_{\text{DSM}}^t(\theta, \eta_t)$ as

$$\eta_t^* = \begin{cases} \mathbb{E}_{q_t(x_t)}\left[\boldsymbol{s}_\theta^t(x_t)\right], & \text{score prediction;} \\ \mathbb{E}_{q_t(x_t)}\left[\boldsymbol{x}_\theta^t(x_t)\right] - \mathbb{E}_{q_0(x_0)}\left[x_0\right], & \text{data prediction;} \\ \mathbb{E}_{q_t(x_t)}\left[\boldsymbol{\epsilon}_\theta^t(x_t)\right], & \text{noise prediction;} \\ \mathbb{E}_{q_t(x_t)}\left[\boldsymbol{v}_\theta^t(x_t)\right] + \sigma_t\mathbb{E}_{q_0(x_0)}\left[x_0\right], & \text{velocity prediction.} \end{cases} \quad (26)$$

Taking $\eta_t^*$ into $\mathcal{J}_{\text{DSM}}^t(\theta, \eta_t)$ we can obtain the gap

$$\mathcal{J}_{\text{DSM}}^t(\theta) - \mathcal{J}_{\text{DSM}}^t(\theta, \eta_t^*) = \begin{cases} \frac{1}{2}\|\mathbb{E}_{q_t(x_t)}\left[\boldsymbol{s}_\theta^t(x_t)\right]\|_2^2, & \text{score prediction;} \\ \frac{\alpha_t^2}{2\sigma_t^4}\|\mathbb{E}_{q_t(x_t)}\left[\boldsymbol{x}_\theta^t(x_t)\right] - \mathbb{E}_{q_0(x_0)}\left[x_0\right]\|_2^2, & \text{data prediction;} \\ \frac{1}{2\sigma_t^2}\|\mathbb{E}_{q_t(x_t)}\left[\boldsymbol{\epsilon}_\theta^t(x_t)\right]\|_2^2, & \text{noise prediction;} \\ \frac{\alpha_t^2}{2\sigma_t^2}\|\mathbb{E}_{q_t(x_t)}\left[\boldsymbol{v}_\theta^t(x_t)\right] + \sigma_t\mathbb{E}_{q_0(x_0)}\left[x_0\right]\|_2^2, & \text{velocity prediction.} \end{cases} \quad (27)$$

## C  Visualization of the generations

We further show generated images in Figure 5 to double confirm the efficacy of our calibration method. Our calibration could help to reduce ambiguous generations on both CIFAR-10 and CelebA.

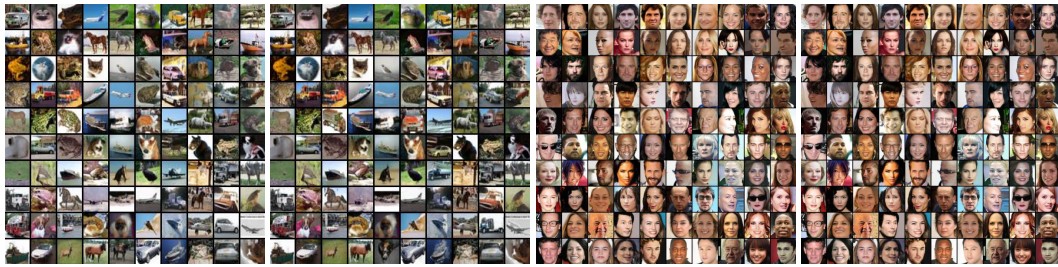

(a) CIFAR-10, w/ calibration     (b) CIFAR-10, w/o calibration     (c) CelebA, w/ calibration     (d) CelebA, w/o calibration

Figure 5: Unconditional generation results on CIFAR-10 and CelebA using models from [17] and [41] respectively. The number of sampling steps is 20 based on the results in Tables 1 and 2.

