# OpenReview forum: "On Calibrating Diffusion Probabilistic Models"
_NeurIPS.cc/2023/Conference — NeurIPS 2023 poster_

### Official Review · Reviewer_DT4C · 2023-07-05

**Soundness:** 3 good
**Presentation:** 3 good
**Contribution:** 2 fair
**Rating:** 5
**Confidence:** 3

**Summary:**

This paper investigates calibrating diffusion models. They notice that the expected data score equals zero. Since models not usually do not learn this correctly, they introduce a calibration term to subtract from a learned score. The new objective now includes the expected model score and performs better compared to the uncalibrated models.

**Strengths:**

The paper is well written and the method is presented clearly. The theoretical results are convincing. The main appeal of the method is its simplicity. The solver part of the results is solid.

**Weaknesses:**

Since the focus is on post-processing, the method is best suited for discrete time diffusion (like DDPM). Otherwise, it is unclear how the expected values are saved for continuous t. Line 221 mentions the selection of t depends on the sampling schedule of the solver, but the solver then cannot be adaptive. The alternative of learning the network as in eq. 14 would solve this but seems overly complicated. As can be seen from Fig 1 the lines are simple so fitting some simpler model would make more sense.

The results are showing the number of solver evaluations is lower and the likelihood is better. Likelihood is evaluated in a weird way. The only thing that is shown is that the expected learned score is not zero for "uncalibrated" models which is somewhat tautological. In this case, some other relevant metric like FID would be better.

There is no empirical demonstration that the diffusion models are uncalibrated, in a similar way that confidence-accuracy diagrams show this for classification. One could learn some simple synthetic dataset from a known distribution.

It is not clear why this would be a relevant task, as presented. I assume guidance would further uncalibrate the models but it is used for better samples. Perhaps the image domain is not the best choice here.

Eq. 13 is presented as a fact, whereas it seems this is to be demonstrated.

The main issue is the limited motivation for improving generative models on likelihood and a bad choice of experimental setup. Something other than images could solve both of these issues.

**Questions:**

- Why is FID going up with the number of samples in Table 3 in both columns?

- Can you incorporate this loss somehow during training so that the final model is calibrated?

- Does this hold for any data distribution?

- Could using 20 and 50 diffusion steps for generation be too little?

**Limitations:**

Limitations are addressed.

---

> ### Author Rebuttal · Authors · 2023-08-07
>
> Thank you for your valuable review and suggestions, we have uploaded a rebuttal PDF.
>
> ***W1: It is unclear how the expected values are saved for continuous t and adaptive sampling schedules***
>
> In our work, we present two methods for implementing calibration: post-training computation and dynamical recording.
>
> Post-training computation can be applied to both discrete and continuous timestep $t$ for non-adaptive sampling schedules. For adaptive sampling schedules, directly applying post-training computation may necessitate additional tricks such as interpolation.
>
> Dynamical recording can thus be used for both discrete/continuous timesteps and non-adaptive/adaptive sampling schedules. The learning framework in Eq. (14) is simply a regression between the recording network and the score outputs, which can be easily implemented with a few lines of code. Furthermore, because we use a shallow MLP for recording (described in Lines 312-322), which is relatively lightweight in comparison to diffusion models, the extra computational and memory costs of dynamical recording are less than $1\\%$.
>
> ***W2: Likelihood evaluation and showing results on other relevant metrics like FID would be better***
>
> As indicated by Eq. (11), the entire area under the curves (i.e., integral w.r.t. timestep $t$) in Figure 1 is counted as the likelihood improvements led by our calibration, for both SDE and ODE solvers.
>
> In **Table A** of the rebuttal PDF, we assess sample quality using FID and other relevant metrics including sFID, inception score (IS), precision and recall. As can be seen, our calibration consistently improves sample quality under different metrics, and we will present full results in the revision.
>
> ***W3: No empirical demonstration that diffusion models are uncalibrated, in a similar way as confidence-accuracy diagrams***
>
> Although we use the same term, there is no direct relationship between the confidence-accuracy calibration studied in discriminative learning and the generative calibration studied in our paper. Intuitively, we observe some essential properties that should hold for any data scores (e.g., the expected data scores should be zero) and define a diffusion model as `uncalibrated’ if its learned model scores deviate from these essential properties. The mechanism of our calibration, such as Eq. (7), is more similar to variance reduction techniques (though not the same; we are more like DSM-loss reduction), which exploit existing observations to obtain better (score) estimators.
>
> ***W4: The main issue is the limited motivation for improving generative models on likelihood***
>
> Improving model likelihood has long been a primary goal in generative learning; for example, generative models such as variational autoencoders (VAEs), energy-based models (EBMs), normalizing flows, and autoregressive models are all trained by maximizing likelihood.
>
> Improving model likelihood has also been well-motivated and widely studied in diffusion models [1,2,3,4,5]. Model likelihood is a principled metric that indicates how well a model distribution is learned towards the data distribution (under KL divergence) and can be practically applied for data compression and density estimation.
>
> ***Q1: Why is FID going up with the number of samples in Table 3 in both columns?***
>
> This phenomenon may be related to ODE solvers. It is found that using more neural function evaluations (NFE) for ODE solvers may increase the FID score [6,7], implying that an overly accurate estimation of the score may not necessarily decrease the FID score. As a result, in Table 3, using more training/generated data to obtain a more accurate estimate of the score mean does not result in a lower FID score. We will conduct ablation studies to further investigate this phenomenon in the revision.
>
> ***Q2: Can you incorporate this loss during training so that the final model is calibrated?***
>
> Yes, we could intuitively incorporate a loss of, say, $\\|\\mathbb{E}\_{q\_{t}(x\_{t})}[\\boldsymbol{\\epsilon}\^{t}\_{\\theta}(x\_{t})]\\|$ during training. However, since the expectation operator $\\mathbb{E}\_{q\_{t}(x\_{t})}$ is inside the norm operator $\\|\\cdot\\|$ (the norm operator is convex), a potential concern is that we can only obtain a biased estimation of $\\|\\mathbb{E}\_{q\_{t}(x\_{t})}[\\boldsymbol{\\epsilon}\^{t}\_{\\theta}(x\_{t})]\\|$ during training using samples in a mini-batch, and the bias could be large for small mini-batch sizes. Post-training computation and dynamical recording, on the other hand, can use more data samples to obtain asymptotically unbiased estimates.
>
> ***Q3: Does this hold for any data distribution?***
>
> Yes, our conclusions apply to any data distribution considered in the diffusion model literature (satisfying mild regularity conditions such as $q\_{0}(x_{0})\\rightarrow 0$ when $x\_{0}\\rightarrow \\infty$, as assumed by default in [8, 9]).
>
>
> ***Q4: Could using 20 and 50 diffusion steps for generation be too little?***
>
> Existing ODE solvers can generate high-quality images with $20 \\sim 50$ (or even $10 \\sim 20$) diffusion steps [6,7].
>
>
> ***References:*** \
> [1] Kingma et al. Variational Diffusion Models. NeurIPS 2021 \
> [2] Song et al. Maximum Likelihood Training of Score-Based Diffusion Models. NeurIPS 2021 \
> [3] Huang et al. A Variational Perspective on Diffusion-Based Generative Models and Score Matching. NeurIPS 2021 \
> [4] Vahdat et al. Score-based Generative Modeling in Latent Space. NeurIPS 2021 \
> [5] Lu et al. Maximum Likelihood Training for Score-Based Diffusion Odes by High Order Denoising Score Matching. ICML 2022 \
> [6] Lu et al. DPM-Solver: A Fast ODE Solver for Diffusion Probabilistic Model Sampling in Around 10 Steps. NeurIPS 2022 \
> [7] Karras et al. Elucidating the Design Space of Diffusion-Based Generative Models. NeurIPS 2022 \
> [8] Ho et al. Denoising Diffusion Probabilistic Models. NeurIPS 2020 \
> [9] Song et al. Score-Based Generative Modeling through Stochastic Differential Equations. ICLR 2021

---

> > ### Comment · Reviewer_DT4C · 2023-08-15
> >
> > Thank you for your detailed response.
> >
> > Many of my questions are adequately addressed. I still think Q1 should be investigated, W3 and Q3 can be demonstrated on some "tabular" data. Your images in the attached pdf (Figure A) are not very convincing. Images 1, 2, 6, 8 offer not improvement in my opinion.
> >
> > However, I am satisfied with the rest of the response and the numerical results so I would not mind acceptance. I'm raising my score accordingly.

---

> > > ### Author Response · Authors · 2023-08-16
> > > **Thank you for your feedback**
> > >
> > > Thank you for your comments and raised score. We will conduct ablation studies to further investigate Q1, and attempt to empirically validate our calibration on data modalities other than vision (e.g., tabular data). We will also provide additional visualization to demonstrate the effectiveness of our calibration.

---

### Official Review · Reviewer_hZuU · 2023-07-06

**Soundness:** 4 excellent
**Presentation:** 3 good
**Contribution:** 3 good
**Rating:** 7
**Confidence:** 4

**Summary:**

The paper proposes a simple procedure to improve the calibration of pre-trained diffusion models.

Diffusion models have achieved strong practical performance, but its score estimation is often seen as a black box. The paper sheds some light on the issue, by detecting the inherent lack of calibration of the current state-of-the-art models, and proposing a simple procedure that reliably drives up the lower bound for model likelihood. Empirical evaluations showcase the efficacy of the method.

**Strengths:**

The paper discusses the problem of calibration in diffusion models, a question long overdue in this area of research: while diffusion models have achieved eye-catching generative results, little is known about whether the score function is a loyal depiction of the ground truth, or even a gradient field in itself.

The paper gives theoretically sound justifications of by framing the evolution of the score function as a martingale, and proposes a simple yet empirically quite effective way that adjust pre-trained diffusion models closer to the underlying score function. The overall writing and presentation of the paper is sound.

**Weaknesses:**

I think Section 3.4 is unsatisfactory, as I think the proper conclusion should be that the uncalibrated DPM training objective do not inherently minimize the mean of the expected predicted noises. My 2 more specific comments are listed below.

- The logic between the explanations in 3.4 and Figure 2 seems confounding. I have understood the main points made in the paper after several reads, so I think the paper could adopt an easier to follow explanation. I believe the author's main point is (i) the optimal score attainable via diffusion modeling is capped by finite training data, so while the optimal score $\nabla \log q_t(x_t;\mathbb{D})$ has expectation zero with respect to the marginal distribution conditioned on data, but it does not likely have expectation zero when given the entirety of $q_t$; and (ii) state-of-the-art diffusion models do not inherently minimize $\lVert\mathbb{E}_{q_t} s_\theta^t \rVert$. I believe those two points are quite separate but lines 188-191 present it as one singular point, which seems as if the paper treats the diffusion modeling as the gold standard of learning the scores, and that even the true score carries a bias in a finite-data setting.
- I admit that the finite data bias can present a problem, but it is not what the calibration trick is for. Frankly speaking, the calibration trick does not manufacture new training data just because it is more calibrated, but only pushes the score estimate _towards_ the optimal score estimate attainable in a finite data regime, i.e., $\nabla \log q_t(x_t;\mathbb{D})$. It is with this argument that I believe the calibration trick will attempt to modify the score estimate even when presented with the ground truth $\nabla \log q_t(x_t)$. Therefore, I believe the main point is not about dataset bias, but that common score matching objectives do not take into consideration the minimization of $\lVert\mathbb{E}_{q_t} s_\theta^t \rVert$.

**Questions:**

None.

**Limitations:**

I believe that the authors can make a few improvements in its presentation of empirical results. I understand that page limit prohibits the addition of more content, but I believe that some plots deserve to be in the main body of text.
- Figure 1 showcases the potential lack of calibration for pre-trained DPMs, but does not show the expectation of predicted noise _after_ calibration. I believe that one can show the effectiveness of calibration by comparing the 2 images and discovering the calibrated model has a much lower expected value of the predicted noise.
- The paper could benefit from also presenting some generated images as it shows any improvements in the most straightforward way. Currently there is a figure in the appendix about how calibration "reduce ambiguous generations", but there are too many examples to see a measurable difference. I believe a smaller plot highlighting the said ambiguous generations deserves a spot in the main text.

---

> ### Author Rebuttal · Authors · 2023-08-07
>
> Thank you for your supportive review and suggestions, we have uploaded a rebuttal PDF.
>
> ***W1: About Section 3.4***
>
> Thank you for your insightful analyses, which are greatly helpful to us. Actually, we conducted preliminary trials on incorporating a loss of $\\|\\mathbb{E}\_{q\_{t}(x\_{t})}[\\boldsymbol{\\epsilon}\^{t}\_{\\theta}(x\_{t})]\\|\^{2}$ during training. However, since the expectation operator $\\mathbb{E}\_{q\_{t}(x\_{t})}$ is inside the norm operator $\\|\\cdot\\|$ (the norm operator is convex), we can only obtain a biased estimation of $\\|\\mathbb{E}\_{q\_{t}(x\_{t})}[\\boldsymbol{\\epsilon}\^{t}\_{\\theta}(x\_{t})]\\|$ during training using samples in a mini-batch, and the bias could be large for small mini-batch sizes. In contrast, our calibration using either post-training computation or dynamical recording can exploit more data samples to obtain asymptotically unbiased estimates of $\\mathbb{E}\_{q\_{t}(x\_{t})}[\\boldsymbol{\\epsilon}\^{t}\_{\\theta}(x\_{t})]$ (as stated in Line 204 for dynamical recording). According to your suggestions, we will better clarify the claims of Section 3.4 and Figure 2 in the revision.
>
>
> ***Limitations***
>
> As to Figure 1, the expectation of predicted noise after calibration is zero as $\\|\\mathbb{E}\_{q\_{t}(x\_{t})}[\\boldsymbol{\\epsilon}\^{t}\_{\\theta}(x\_{t})-\\mathbb{E}\_{q\_{t}(x\_{t})}[\\boldsymbol{\\epsilon}\^{t}\_{\\theta}(x\_{t})]]\\|=0$ for noise prediction and $\\|\\mathbb{E}\_{q\_{t}(x\_{t})}[\\boldsymbol{s}\^{t}\_{\\theta}(x\_{t})-\\mathbb{E}\_{q\_{t}(x\_{t})}[\\boldsymbol{s}\^{t}\_{\\theta}(x\_{t})]]\\|=0$ for score prediction. Therefore, as indicated by Eq. (11), the entire area under the curves in Figure 1 is counted as the likelihood improvements led by our calibration.
>
>
> In **Figure A** of the rebuttal PDF, we provide examples demonstrating that our calibration could reduce ambiguous generations, such as eliminating generations that resemble both horse and dog. Intuitively, uncalibrated scores will contain redundant information from all classes and lead to the generation of ambiguous features.

---

> > ### Comment · Reviewer_hZuU · 2023-08-10
> > **Post-rebuttal response**
> >
> > I thank the author for explaining points I laid out in the reviews: I maintain the same score assessment for the paper as I already recommend acceptance.

---

> > > ### Author Response · Authors · 2023-08-10
> > > **Thank you for your feedback**
> > >
> > > We appreciate your detailed comments, especially the insightful suggestions about Section 3.4. We will incorporate them into the final revision of our paper. Thank you again!

---

### Official Review · Reviewer_LmXp · 2023-07-06

**Soundness:** 4 excellent
**Presentation:** 3 good
**Contribution:** 3 good
**Rating:** 7
**Confidence:** 3

**Summary:**

This paper introduces a general calibration technique for diffusion probabilistic models (DPMs). The authors derive a time-dependent calibration term that is independent of any particular input under different model parameterizations. This calibration term can be computed in advance and repeatedly used for sampling. The experimental results demonstrate the effectiveness of the proposed calibration method with a noticeable reduction of FIDs when taking 15-40 NFEs.

**Strengths:**

The proposed calibration method in this paper is novel, theoretically sound, and practically useful in that it in principle can be applied to all kinds of DPMs to improve sampling with little computational overhead. The authors provide a performance analysis with varying numbers of samples for approximating the calibration term using the MC method. The FID results suggest that 20K of MC samples are sufficient to estimate the calibration term and reach the best quality. In this sense, the pre-sampling computation seems not to be too expensive. Overall, this approach appears to be general and easy to implement. Therefore, I believe this paper is significant and worth acceptance in NeurIPS.

**Weaknesses:**

Although the calibration term is not $x_t$-dependent, it depends on time ($t$) and the model parameters ($\theta$). I am therefore concerned about the stability of the proposed approach. From Figure 1, we can basically confirm that the range of calibration terms drastically varies over different data sets. In this sense, if we apply the proposed approach to a new data set, there are at least three dynamic factors -- $t$, $\theta$, and data distribution $q_0(x_0)$ that could affect the performance of this calibration method. In regard to this, I suggest the author conduct more validation experiments to prove the stability and consistency of the proposed method over different time steps, models, and datasets.

**Questions:**

What are the FID results of AFHQv2 64×64, FFHQ 64×64, and ImageNet 64×64? Following my concerns on stability, in the main paper, the authors state that they have conducted experiments on these datasets based on pre-trained models considering their top performance. However, I cannot find these numbers in the main paper and in the appendix. Why do you choose to only present CIFAR-10 and CelebA 64x64 in the main paper? I believe that an improvement on the ImageNet 64×64 is more convincing. I suggest the authors keep the FIDs of all these datasets in the main paper and compare them to the original DPMs.

**Limitations:**

The impact of this paper may be limited by the lack of comparison against other existing methods. I tend to categorize this paper into research that improves pre-trained DPMs generally at inference time. Along this direction of research, there are several relevant works, which I believe the author could easily compare to, i.e., (not trainable) PNDM [1],  Analytic-DPM [2], and (trainable) BDDM [3].

[1] Liu, Luping, et al. "Pseudo Numerical Methods for Diffusion Models on Manifolds."
[2] Bao, Fan, et al. "Analytic-DPM: an Analytic Estimate of the Optimal Reverse Variance in Diffusion Probabilistic Models."
[3] Lam, Max WY, et al. "Bilateral Denoising Diffusion Models."

---

> ### Author Rebuttal · Authors · 2023-08-07
>
> Thank you for your supportive review and suggestions, we have uploaded a rebuttal PDF.
>
> ***W1 & Q1: The stability of the proposed approach and FID results of AFHQv2 64×64, FFHQ 64×64, and ImageNet 64×64***
>
> Indeed, the gain of our calibration trick is proportional to the degree of `uncalibration’ for a diffusion model, or how far the uncalibrated learned scores deviate from the essential properties (e.g., the expected data scores should be zero). We will conduct more validation experiments to demonstrate the stability and consistency of our calibration.
>
> **Table B** of the rebuttal PDF contains the FID results for AFHQv2 64×64, FFHQ 64×64, and ImageNet 64×64. To reduce computational burden, we did not run full FID experiments on these datasets in the original main paper. In the revision, we will provide full FID results on AFHQv2 64×64, FFHQ 64×64, and ImageNet 64×64.
>
> ***Limitations***
>
> Thank you for your constructive suggestions. Our calibration is conceptually compatible with these advanced samplers/solvers [1,2,3], because we focus on improving model score estimation and these methods focus on how to efficiently utilize model scores for better/faster sampling. In the revision, we will conduct empirical studies to ensure that our calibration is compatible with these methods.
>
> ***References:*** \
> [1] Liu et al. Pseudo Numerical Methods for Diffusion Models on Manifolds. ICLR 2022 \
> [2] Bao et al. Analytic-DPM: an Analytic Estimate of the Optimal Reverse Variance in Diffusion Probabilistic Models. ICLR 2022 \
> [3] Lam et al. Bilateral Denoising Diffusion Models. NeurIPS 2021

---

> > ### Comment · Reviewer_LmXp · 2023-08-11
> > **Response to the authors**
> >
> > Thanks the authors for a detailed response and adding new experimental results. I am convinced that the proposed calibration trick is a general method applicable to diffusion models. I will keep my rating, leaning to an acceptance.

---

> > > ### Author Response · Authors · 2023-08-11
> > > **Thank you for your feedback**
> > >
> > > We appreciate your detailed comments and suggestions. We will polish our paper further and incorporate new results into the final revision. Thank you again!

---

### Official Review · Reviewer_pqYH · 2023-07-06

**Soundness:** 2 fair
**Presentation:** 3 good
**Contribution:** 2 fair
**Rating:** 6
**Confidence:** 3

**Summary:**

This paper presents a simple way to calibrate an arbitrary pretrained diffusion probabilistic model (DPM), which can reduce the score matching loss and increase the lower bounds of model likelihood. The authors observe that the stochastic reverse process of data scores is a martingale, from which concentration bounds and the optional stopping theorem for data scores can be derived. The proposed calibration method is easy to follow and can be performed only once, resulting in models that can be used repeatedly for sampling. The premise is that DPMs are by default uncalibrated (which they show experimentally), and that calibration can result in improved generation. This again is showed experimentally using the FID metric.

—-
Raised score from 4 to 6 post-rebuttal. On the whole it seems like the method can have some positive impacts - mostly in terms of model likelihood (with subsequent downstream benefits), and sometimes in terms of the quality of generated images.

**Strengths:**

- clear and concise presentation of the proposed calibration method
- empirical validation of the method on multiple datasets,
- derivation of concentration bounds and the optional stopping theorem for data scores

**Weaknesses:**

This biggest weakness is solely relying on model likelihood and the FID score for measuring generative performance. FID (Heusel et al., 2017) calculates the Wasserstein-2 (a.k.a Fréchet) distance between multivariate Gaussians fitted to the embedding space of the Inception-v3 network of generated and real image. A major drawback with FID is its high bias. The sample size to calculate FID has to be large enough (usually above 50K). Smaller sample sizes can lead to over-estimation of the actual FID. This can clearly bee seen in Table 3.

It's not a surprise that the model likelihood improves (since this is directly linked to calibration), but the link between model likelihood and generation quality is not at all clear.

**Questions:**

In the supplementary material, some example generated images are given with/without calibration. I zoomed right in, but I struggled to see any difference between pairs of images - is there anything that demonstrates that calibrating improves matters?

**Limitations:**

Yes, the authors have partially addressed the limitations in that they acknowledged that the model likelihood comes down but not necessarily FID, although they don't answer the question of whether it improves generation quality! Also there is a scaling issue with the method that is identified.

---

> ### Author Rebuttal · Authors · 2023-08-07
>
> Thank you for your valuable review and suggestions, we have uploaded a rebuttal PDF.
>
> ***W1: Solely relying on model likelihood and the FID score for measuring generative performance***
>
> First, we need to clarify that Table 3 is an ablation study on *the number of samples used to estimate the calibration term*, rather than the number of samples used to calculate the FID score. As stated in Section 4.1 (Lines 217-218), we use 50K samples to calculate the FID score for all experiments in our paper.
>
> In **Table A** of the rebuttal PDF, we assess sample quality using FID and other performance metrics including sFID, inception score (IS), precision and recall. As can be seen, our calibration consistently improves sample quality under different metrics, and we will present full results in the revision.
>
> ***W2: The link between model likelihood and generation quality is not at all clear***
>
> There have been several works discussing the link between model likelihood and generation quality in diffusion models [1,2,3,4,5], and their observations are that there is usually a trade-off between model likelihood and generation quality (i.e., improving model likelihood may degrade generation quality, and vice versa). In contrast, our calibration could improve both model likelihood and generation quality at the same time.
>
> ***Q1: Is there anything that demonstrates that calibrating improves matters?***
>
> In addition to the quantitative improvements in model likelihood and various generative metrics (FID, sFID, IS, precision and recall), we show in **Figure A** of the rebuttal PDF how our calibration can reduce ambiguous generations, such as eliminating generations that resemble both horse and dog. Intuitively, uncalibrated scores will contain redundant information from all classes and lead to the generation of ambiguous features.
>
>
> ***References:*** \
> [1] Kingma et al. Variational Diffusion Models. NeurIPS 2021 \
> [2] Song et al. Maximum Likelihood Training of Score-Based Diffusion Models. NeurIPS 2021 \
> [3] Huang et al. A Variational Perspective on Diffusion-Based Generative Models and Score Matching. NeurIPS 2021 \
> [4] Vahdat et al. Score-based Generative Modeling in Latent Space. NeurIPS 2021 \
> [5] Lu et al. Maximum Likelihood Training for Score-Based Diffusion Odes by High Order Denoising Score Matching. ICML 2022

---

> > ### Comment · Reviewer_pqYH · 2023-08-13
> >
> > Thanks for the responses and clarifications.
> >
> > Ultimately it comes down to those images. I concur that 9 of them do look better the calibration (the car image I don’t really think is much different). Perhaps there are also other images that look better without calibration (i.e. this is one-sided)? Perhaps the differences would be more obvious on higher dimensional images? I think the jury is still out that calibration is really useful here.

---

> > > ### Author Response · Authors · 2023-08-13
> > > **Thank you for your feedback**
> > >
> > > Thank you for the valuable feedback.
> > >
> > > The phenomenon shown in Figure A is NOT one-sided. According to our observations, the generated images are either *calibration looks better* or *small visually distinguishable difference before/after calibration*. We can hardly find the image where without calibration looks better. Following your suggestions, we will provide more visual examples on higher dimensional images such as ImageNet 64×64 in the revision.
> > >
> > > Beyond visualization, higher model likelihood can directly benefit data compression and density estimation, whereas improved image quality (as measured by quantitative metrics) can benefit downstream tasks such as image editing/customization, semi-supervised learning, and improving adversarial robustness.

---

> > > > ### Comment · Reviewer_pqYH · 2023-08-13
> > > >
> > > > Ok thanks this helps. I’ll raise my rating accordingly.
> > > > Since you mention that it can help compression, it occurs that this is something else that is measurable that could help strengthen the case (unless I miss that you already had metrics around this).

---

> > > > > ### Author Response · Authors · 2023-08-13
> > > > > **Thank you for the further comments**
> > > > >
> > > > > Thank you for the further comments and raised rating.
> > > > >
> > > > > Likelihood-based generative models such as VAEs, autoregressive models, and normalizing flows have been studied for lossless compression, where the optimal codelength is closely related to the negative log-likelihood of generative models [a,b]. More recently, Section 6.3 of Kingma et al. [c] discuss how likelihood of diffusion models can be turned into lossless compression algorithm using bits-back coding. We will add relevant discussion in the final revision.
> > > > >
> > > > > [a] Townsend et al. Practical Lossless Compression with Latent Variables Using Bits Back Coding. ICLR 2019 \
> > > > > [b] Ho et al. Compression with Flows via Local Bits-Back Coding. NeurIPS 2019 \
> > > > > [c] Kingma et al. Variational Diffusion Models. NeurIPS 2021

---

### Official Review · Reviewer_2qSH · 2023-07-08

**Soundness:** 4 excellent
**Presentation:** 3 good
**Contribution:** 3 good
**Rating:** 7
**Confidence:** 4

**Summary:**

The paper makes an observation regarding the reverse process of diffusion probabilistic model, noting that the data score term  is a martingale with respect to this process. A key contribution of the paper is a theorem to this effect and the associated proof. Leveraging this observation, the authors propose a calibration technique: one can calibrate a pretrained diffusion model at any time step by subtracting its expectation. Experimental results in the paper demonstrate that the calibrated score model achieves lower values of score matching objectives. Furthermore, the paper provides evidence that the calibrated score model yields higher evidence lower bounds. Lastly, the paper explains that similar conclusions hold true for the conditional score model.

**Strengths:**

S1. The paper makes a novel and significant observation concerning the martingale nature of the data score term in probabilistic diffusion models with respect to the reverse process.

S2. The paper is clearly written, well-organized, and provides a key theorem and proof that are presented in a readily-followed fashion.

S3.  The innovative technique for calibration is soundly based on theoretically-derived principles, is relatively simple to compute, and achieves an experimental performance improvement. The method is extended to the conditional setting.

S4. The paper provides visualizations of the expected predicted noises and associated discussion that helps the reader understand the inductive bias. The additional experimental analysis is a welcome addition (sensitivity to the number of training samples, performance using generated samples, dynamic recording performance).


**Weaknesses:**

W1. Although the proposed method achieves an improvement in terms of model likelihood, the experiments are less convincing that there is a meaningful practical improvement in terms of generated samples. There is an improvement in FID, but the sample images are presented in the appendix in such a way that it is almost impossible to discern a difference between the calibrated and uncalibrated models.

W2. The reported experiments only assess sample quality using FID. Multiple papers have demonstrated how this metric can be misleading in some circumstances and have proposed alternatives, e.g., precision and recall metrics, that can be used in conjunction with FID to provide a clearer and more complete assessment of the capabilities of a generative model.

W3. There is very little discussion of computational and memory requirements for the calibration. The paper could benefit from a much clearer discussion of the costs that are incurred in order to obtain the calibration benefit.


**Questions:**

Q1. Can the authors provide a clearer commentary about the claimed subjective quality improvement. The generated images in Appendix C are extremely small. There is almost no discussion beyond “Our calibration could help to reduce ambiguous generations” – this is almost impossible for a reader to verify with the way the images are presented and it is not clear exactly what is meant.

Q2. Could the authors comment on how the method performs for other performance metrics for generative models?

Q3. What is the computational and memory cost (particulary in comparison to the original model)? i.e., does it increase the computational overhead by 30 percent? Is the memory cost an additional 20 percent?


**Limitations:**

The paper provides a very brief discussion of the limitations. This is definitely useful, and the authors acknowledge some of the key limitations. On the other hand, it would strengthen the paper if the supplementary material contained a more in-depth discussion. In particular, additional discussion concerning overhead and whether the proposed method translates to genuine practical benefits would be welcome. The section could discuss the challenge of evaluating generative models and suggest potential research avenues fhat might help establish whether the calibration improvement translates to more applied settings.

---

> ### Author Rebuttal · Authors · 2023-08-07
>
> Thank you for your supportive review and suggestions, we have uploaded a rebuttal PDF.
>
> ***W1 & Q1: Providing a clearer commentary on the improvement of image quality***
>
> In **Figure A** of the rebuttal PDF, we provide examples demonstrating that our calibration could reduce ambiguous generations, such as eliminating generations that resemble both horse and dog. Intuitively, uncalibrated scores will contain redundant information from all classes and lead to the generation of ambiguous features.
>
> ***W2 & Q2: Generative performance under other metrics***
>
> In **Table A** of the rebuttal PDF, we assess sample quality using FID and other performance metrics including sFID, inception score (IS), precision and recall. As can be seen, our calibration consistently improves sample quality under different metrics, and we will present full results in the revision.
>
> ***W3 & Q3: Computational and memory costs***
>
> We provide two methods for implementing calibration in our work: post-training computation and dynamical recording.
>
> For post-training computation, we compute the score mean for each inference timestep (e.g., each timestep utilized by DPM-Solver) and restore them for reuse. Therefore, the amortized computational cost for calibration is precisely $\\frac{\\mathcal{M}}{\\mathcal{N}}$, where $\\mathcal{M}$ is the number of samples used to calculate each score mean, and $\\mathcal{N}$ is the number of new samples generated during inference. If we use $\\mathcal{M}=20,000$ samples to calculate each score mean as ablated in Table 3, and we generate $\\mathcal{N}=100,000$ new samples during inference, the amortized computational cost will be $20\\%$; if we generate more samples, such as $\\mathcal{N}=1,000,000$, the amortized computational cost will be $2\\%$. As to the memory cost, since we first calculate score means and then restore them as detached tensors, they consume negligible inference memory during generation.
>
> For dynamical recording, both the extra computational and memory costs are less than $1\\%$, because we use a shallow MLP for recording, which is relatively lightweight in comparison to diffusion models. In the revision, we will provide more detailed analyses of computational and memory costs.
>
> ***Limitations***
>
> Thank you for the instructive suggestions. Dynamical recording could significantly reduce the computational and memory overheads of calibration, while a calibrated diffusion model could potentially benefit downstream applications such as image editing/customization, semi-supervised learning, and improving adversarial robustness. In addition to model likelihood and quality metrics, we can evaluate a diffusion model by its degree of `uncalibration’, namely, how far the learned scores deviate from the essential properties (e.g., the expected data scores should be zero). In the revision, we will have a more in-depth discussion.

---

> > ### Comment · Reviewer_2qSH · 2023-08-12
> >
> > I thank the authors for their detailed response. It has resolved my questions. Since I already recommended acceptance, I have retained my original ranking.

---

> > > ### Author Response · Authors · 2023-08-13
> > > **Thank you for your feedback**
> > >
> > > We appreciate your detailed comments and suggestions. Our final revision will include new performance results as well as details on computational/memory costs. Thank you again!

---

### Author Rebuttal · Authors · 2023-08-07

We thank all reviewers for their constructive feedback, and we have responded to each reviewer individually. We have also uploaded a rebuttal PDF that includes:

- **Table A**: Assessing sample quality using FID and other performance metrics including sFID, inception score (IS), precision and recall.

- **Table B**: FID results for AFHQv2 64×64, FFHQ 64×64, and ImageNet 64×64.

- **Figure A**: Selected examples demonstrating that our calibration could reduce ambiguous generations.

---

### Decision · Program_Chairs · 2023-09-21

**Decision:**

Accept (poster)

**Comment:**

The authors propose a novel calibration technique wherein a pretrained diffusion model using a time-dependent term to enforce zero-mean of the score function.
This insight stems from the fact that it holds for the true score function of the corresponding diffusion process.

All reviewers and I agree that this papers makes a good contribution: the proposed correction term is easy to implement and the authors succeeds in justifying its use. In addition, method's advantages are well-supported through numerical experiments.

As you proceed to prepare the final version of your work for submission, kindly take into account the feedback provided by the reviewers in order to enhance the quality and clarity of your paper.